# How to Solve Contextual Goal-Oriented Problems with Offline Datasets?

**Ying Fan**[1], **Jingling Li**[2], **Adith Swaminathan**[3], **Aditya Modi**[3], **Ching-An Cheng**[3]
[1]University of Wisconsin-Madison [2]ByteDance Research [3]Microsoft Research

## Abstract

We present a novel method, Contextual goal-Oriented Data Augmentation (CODA), which uses commonly available unlabeled trajectories and context-goal pairs to solve Contextual Goal-Oriented (CGO) problems. By carefully constructing an action-augmented MDP that is equivalent to the original MDP, CODA creates a fully labeled transition dataset under training contexts without additional approximation error. We conduct a novel theoretical analysis to demonstrate CODA's capability to solve CGO problems in the offline data setup. Empirical results also showcase the effectiveness of CODA, which outperforms other baseline methods across various context-goal relationships of CGO problem. This approach offers a promising direction to solving CGO problems using offline datasets.

## 1 Introduction

Goal-oriented problems [16] are an important class of sequential decision-making problems with widespread applications, ranging from robotics [39], game-playing [12], to logistics [24]. In particular, many real-world goal oriented problems are *contextual*, where the objective of the agent is to reach a goal set communicated by a context. For example, consider instructing a truck operator with the context "Deliver goods to a warehouse in the Bay area". Given such a context and an initial state, it is acceptable to reach any feasible goal (a reachable warehouse location) in the goal set (warehouse locations including non-reachable ones). We call such problems *Contextual Goal-Oriented* (CGO) problems, which form an important special case of contextual Markov Decision Process (MDP) [10].

CGO is a practical setup that includes goal-conditioned reinforcement learning (GCRL) as a special case (the context in GCRL is just the target goal), but in general contexts in CGO problem can be more abstract (like high-level task instructions in the above example) and the relationship between contexts and goals are not known beforehand. CGO problems are challenging because 1) the rewards are sparse as in GCRL and 2) the contexts can be difficult to map into feasible goals. Nevertheless, CGO problem has an important structure that the transition dynamics (e.g., navigating a city road network) are independent of the contexts that specify tasks. Therefore, efficient multitask learning can be achieved by sharing dynamics data across tasks.

In this paper, we study solving for CGO problems in an offline setup. We suppose access to two datasets — an (unlabeled) *dynamics* dataset of trajectories, and a (labeled) *context-goal* dataset containing pairs of contexts and goal examples. Such datasets are commonly available in practice. The typical contextual datasets for imitation learning (IL) (which has pairs of contexts and expert trajectories) is one example, since we can convert the contextual IL data into dynamics data and context-goal pairs. Generally, this setup also covers scenarios where expert trajectories are *not* accessible (e.g., because of diverse contexts and initial states), since it does not assume goal examples to appear in the trajectories or the contexts are readily paired with transitions in expert trajectories. Instead, it allows the dynamics datasets and the context-goal datasets to be independently collected. For example, in robotics, task-agnostic play data can be obtained at scale [22, 34] in an unsupervised manner whereas instruction datasets (e.g., [25]) can provide context-goal pairs. In navigation, self-

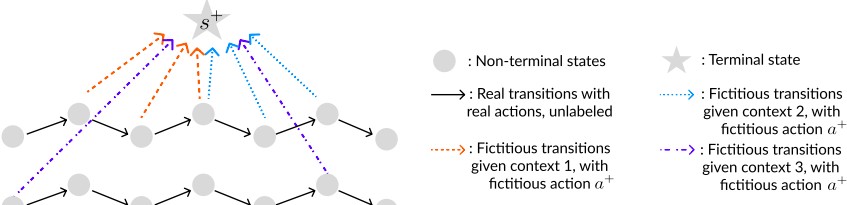

Figure 1: Illustration of CODA: We create fictitious transitions from goal examples to terminal states under the given context in the action-augmented MDP with reward 1, which enables the supervised signal to propagate back to unsupervised transitions via Bellman equation.[1]

driving car trajectories (e.g., [35, 32]) also allow us to learn dynamics whereas landmarks datasets (e.g. [24, 9]) provide context-goal pairs.

While offline CGO problems as described above are common in practical scenarios, to our knowledge, no algorithms have been specifically designed to solve such problems and CGO has not been formally studied yet. Some baseline methods could be easily conceptualized from the literature, but their drawbacks are equally apparent. One intuitive approach is to extend the goal prediction methods in GCRL [26, 27]: given a test context, we can predict a goal and navigate to it using a goal-conditioned policy, where the goal prediction model can be learned from the context-goal dataset and the goal-conditioned policy can be learned from the trajectory dataset. However, the predicted goal might not always be feasible given the initial state since our context-goal dataset is not necessarily paired with transitions. Alternatively, the offline problem could be formulated as a special case of missing label problems [41] and we can learn a context-conditioned reward model to label the unsupervised transitions when paired with contexts as in [14]. However, this approach ignores the goal-oriented nature of the problem and the fact that here only positive data (i.e. goal examples) are available for reward learning, which poses extra significant challenges. CGO can be framed as an offline reinforcement learning (RL) problem with missing labels; However, existing algorithms [42, 14, 21] in family assume access to both positive data (contexts-goal pairs) and negative data (contexts and non-goal examples), whereas only positive data are available here.

In this work, we present the first precise formalization of the CGO setting, and propose a novel Contextual goal-Oriented Data Augmentation (CODA) technique that can provably solve CGO problems subject to natural assumptions on the datasets' quality. The core idea is to **convert the context-goal dataset and the unsupervised dynamics dataset to a fully labeled transition dataset of an equivalent action-augmented MDP**, which circumvents the drawbacks in other baseline methods by fully making use of the CGO structure of the problem. We give a high-level illustration of this idea in Figure 1. In Figure 1, given a randomly sampled context-goal pair from the context-goal dataset, we create fictitious transitions from the corresponding goal example to a fictitious terminal state with a fictitious action and reward 1, and pair with the corresponding context. Also, we label all unsupervised transitions with reward 0 and non-terminal, and pair with the contexts randomly. Combining the two, we then have a fully labeled dataset (of an action-augmented contextual MDP, which this data augmentation and relabeling process effectively creates), making it possible to propagate supervision signals from the context-goal dataset to unsupervised transitions via the Bellman equation. We can then apply any offline RL algorithm based on Bellman updates like CQL [19], IQL [18], PSPI [37], ATAC [4] etc. In comparison with the baseline methods discussed earlier, our method naturally circumvents their intrinsic challenges: 1) CODA directly learns context-conditioned policy and avoids the need to predict goals; 2) CODA effectively uses a fully labeled dataset, avoiding the need to learn a reward model and extra costs from inaccurate reward modeling.

## 2 Related Work

**Offline RL.** Offline RL methods have proven to be effective in goal-oriented problems as it also allows learning a common set of sub-goals/skills [3, 23, 38]. A variety of approaches are used to mitigate the distribution shift between the collected datasets and the trajectories likely to be generated by learned policies: 1) constrain target policies to be close to the dataset distribution [8, 36, 7], 2) incorporate value pessimism for low-coverage or Out-Of-Distribution states and actions [19, 40, 15] and 3) adversarial training via a two-player game [37, 4].

**Offline RL with unlabeled data.** Our CGO setting is a special case of offline RL with unlabeled data, or more broadly the offline policy learning from observations paradigm [21]: There is only a subset of the offline data labeled with rewards (in our setting, that is the contexts dataset, as we don't know which samples in the dynamics dataset are goals.). However, the MAHALO scheme in [21] is much more general than necessary for CGO problems, and we show instead that our CODA scheme has better theoretical guarantees than MAHALO in Section 5. In our experiments, we compare CGO with several offline RL algorithms designed for unlabeled data: UDS [42] where unlabeled data is assigned zero rewards and PDS [14] where a pessimistic reward function is learned from a labeled dataset.

**Goal-conditioned RL (GCRL).** GCRL is a special case of our CGO setting, which has been extensively studied since [16]. There are two critical aspects of GCRL: 1) data relabeling to make better use of available data and 2) learning reusable skills to solve long-horizon problems by chaining sub-goals or skills. On the one hand, hindsight relabeling methods [1, 20] are effective by reusing visited states in the trajectories as successful goal examples. For 2), hierarchical methods for determining sub-goals, and training goal reaching policies have been effective in long-horizon problems [28, 30, 3]. Another key objective of GCRL is goal generalization. Popular strategies include universal value function approximators [29], unsupervised representation learning [26, 28, 11], and pessimism-induced generalization in offline GCRL formulations [38]. Our CGO framing enables both data reuse and goal generalization, by using contextual representations and a reduction to offline RL to combine dynamics and context-goal datasets.

**Data-sharing in RL**   Sharing information across multiple tasks is a promising approach to accelerate learning and to identify transferable features across tasks. In RL, both multi-task and transfer learning settings have been studied under varying assumption on the shared properties and structures of different tasks [43, 33, 2, 5]. For data sharing in CGO, we adopt the contextual MDP formulation [10, 31], which enables knowledge transfer via high-level contextual cues. Prior work on offline RL has also shown the utility of sharing data across tasks: hindsight relabeling and manual skill grouping [17], inverse RL [20], sharing Q-value estimates [41, 30] and reward labeling [42, 14].

## 3  Preliminaries

In this section, we introduce the setup of CGO problems, infinite-horizon formulation for CGO, and the offline learning setup with basic assumptions for our offline dataset.

**CGO Setup**   A Contextual Goal-Oriented (CGO) problem describes a multi-task goal-oriented setting with a *shared* transition kernel. We consider a Markovian CGO problem, defined by the tuple $\mathcal{M} = (\mathcal{S}, \mathcal{A}, P, R, \gamma, \mathcal{C}, d_0)$, where $\mathcal{S}$ is the state space, $\mathcal{A}$ is the action space, $P : \mathcal{S} \times \mathcal{A} \to \Delta(\mathcal{S})$ is the transition kernel, $R : \mathcal{S} \times \mathcal{C} \to \{0, 1\}$ is the sparse reward function, $\gamma \in [0, 1)$ is the discount factor, $\mathcal{C}$ is the context space, and $\Delta$ denotes the space of distributions.

Each context $c \in \mathcal{C}$ specifies a goal-reaching task with a goal set $G_c \subset \mathcal{S}$, and reaching any goal in the goal set $G_c$ is regarded as successful, inducing the reward function $R(s, c) = \mathbb{1}(s \in G_c)$. An episode of a CGO problem starts from an initial state $s_0$ and a context $c$ sampled from $d_0(s_0, c)$, and terminates when the agent reaches the goal set $G_c$. $c$ does not change during the transition; only $s_t$ changes according to $P(s'|s, a)$ and the transition kernel is context-independent.

**Infinite-horizon Formulation for CGO setup**   A fictitious zero-reward absorbing state $s^+ \notin \mathcal{S}$ can translate termination after reaching the goal to an infinite horizon formulation: *whenever the agent enters $G_c$ it transits to $s^+$ in the next step (for all actions) and stays at $s^+$ forever.* This is a standard technique to convert a goal-reaching problem (with a random problem horizon) to an infinite horizon problem. This translation does *not* change the problem, but allows cleaner analyses. We adopt this formulation in the following.

We give details of this infinite-horizon conversion in the following. First, we extend the reward and the dynamics: Let $\bar{\mathcal{S}} = \mathcal{S} \bigcup \{s^+\}$, $\mathcal{X} := \mathcal{S} \times \mathcal{C}$, and $\bar{\mathcal{X}} := \bar{\mathcal{S}} \times \mathcal{C}$. Define $\mathcal{X}^+ := \{x : x = (s, c), s = s^+, c \in \mathcal{C}\}$. With abuse of notation, we define the reward and transition on $\bar{\mathcal{X}}$ as $R(x) = \mathbb{1}(s \in G_c)$

where $x = (s, c)$. The transition kernel $P(x'|x, a) := P(s'|s, c, a)\mathbb{1}(c' = c)$, where

$$P(s'|s, c, a) = \begin{cases} \mathbb{1}(s' = s^+) & \text{if } s \in G_c \text{ or } s = s^+, \\ P(s'|s, a) & \text{otherwise.} \end{cases}$$

Given a policy $\pi : \mathcal{X} \to \Delta(\mathcal{A})$, the state-action value function (i.e., Q function) is $Q^\pi(x, a) := \mathbb{E}_{\pi, P}\left[\sum_{t=0}^\infty \gamma^t R(x)|x_0 = x, a_0 = a\right]$. $V^\pi(x) := Q^\pi(x, \pi)$ is the value function given $\pi$, where $Q(x, \pi) := \mathbb{E}_{a \sim \pi}[Q(x, a)] \in [0, 1]$. The return $J(\pi) = V^\pi(d_0) = Q^\pi(d_0, \pi)$. $\pi^*$ is the optimal policy that maximized $J(\pi)$ and $Q^* := Q^{\pi^*}$, $V^* := V^{\pi^*}$. Let $G$ represent the goal set on $\mathcal{X}$, that is, $G := \{x \in \mathcal{X} : x = (s, c), s \in G_c\}$.

**Offline Learning for CGO**    We aim to solve CGO problems using offline datasets without additional online environment interactions, namely, by offline RL. We identify two types of data that are commonly available: $D_{\text{dyn}} := \{(s, a, s')\}$ is an *unsupervised* dynamics dataset of agent trajectories collected from $P(s'|s, a)$, and $D_{\text{goal}} := \{(c, s) : s \in G_c\}$ is a *supervised* dataset of context-goal pairs, which can be easier to collect than expert trajectories. We suppose that there are two distributions $\mu_{\text{dyn}}(s, a, s')$ and $\mu_{\text{goal}}(s, c)$, where $\mu_{\text{dyn}}(s'|s, a) = P(s'|s, a)$ and $\mu_{\text{goal}}$ has support within $G_c$, i.e., $\mu_{\text{goal}}(s|c) > 0 \Rightarrow s \in G_c$. We assume that $D_{\text{dyn}}$ and $D_{\text{goal}}$ are i.i.d. samples drawn from the distributions $\mu_{\text{dyn}}$ and $\mu_{\text{goal}}$, i.e.,

$$D_{\text{dyn}} = \{(s_i, a_i, s'_i) \sim \mu_{\text{dyn}}\}, D_{\text{goal}} = \{(s_j, c_j) \sim \mu_{\text{goal}}\}.$$

Notice that we do not assume the goal states in $D_{\text{goal}}$ to be in $D_{\text{dyn}}$, thus we cannot always naively pair transitions in $D_{\text{dyn}}$ with contexts in $D_{\text{goal}}$ and assign them with reward 1. To our knowledge, no existing algorithm can provably learn near-optimal $\pi$ using only the positive $D_{\text{goal}}$ data (i.e., without non-goal examples) when combined with $D_{\text{dyn}}$ data.

## 4    Contextual Goal-Oriented Data Augmentation (CODA)

The key idea of CODA is the construction of an *action*-augmented MDP with which the dynamics and context-goal datasets can be combined into a fully labeled offline RL dataset. In the following, we first describe this action-augmented MDP (Section 4.1) and show that it preserves the optimal policies of the original MDP (Appendix A.1). Then we outline a practical algorithm to convert the two datasets of an offline CGO problem into a dataset for this augmented MDP (Section 4.2) such that any generic offline RL algorithm based on Bellman equation can be used as a solver.

### 4.1    Action-Augmented MDP

We propose an action-augmented MDP (shown in Figure 1), which augments the action space of the contextual MDP in Section 3 with *a fictitious action* $a^+ \notin \mathcal{A}$.

Let $\bar{\mathcal{A}} = \mathcal{A} \bigcup \{a^+\}$. We define the reward of this action-augmented MDP to be *action-dependent*: for $x = (s, c) \in \mathcal{X}$, $\bar{R}(x, a) := \mathbb{1}(s \in G_c)\mathbb{1}(a = a^+)$, which means the reward is 1 only if $a^+$ is taken in the goal set, otherwise 0.

We also extend the transition upon taking action $a^+$: $\bar{P}(x'|x, a^+) := \mathbb{1}(s' = s^+)$, and maintain the transition with real actions: $\bar{P}(x'|x, a) := P(s'|s, a)\mathbb{1}(c' = c)$, which means whenever taking $a^+$, the agent would always transit to $s^+$, and the transition remains the same as in the original MDP given real actions. Further, we implement $s^+$ as `terminal = True`.

We define this augmented MDP as $\overline{\mathcal{M}} := (\mathcal{X}, \bar{\mathcal{A}}, \bar{R}, \bar{P}, \gamma)$.

**Policy conversion.**    For a policy $\pi : \mathcal{X} \to \Delta(\mathcal{A})$ in the original MDP, define its extension on $\overline{\mathcal{M}}$:

$$\bar{\pi}(a|x) = \begin{cases} \pi(a|x), & x \notin G, \\ a^+, & \text{otherwise.} \end{cases} \tag{1}$$

**Regret equivalence.**    An observation that comes with the construction is that if a policy is optimal in the original MDP, we can easily use the extension above to create an optimal policy in the augmented one. If a policy is optimal in the augmented MDP, it must take $a^+$ only when $x \in G$ (otherwise the return is lower, due to entering $s^+$ too early), thus we can revert this optimal policy of the augmented MDP to find an optimal policy in the original MDP without changing its behavior and performance. We stated this property below; details can be found as Lemma A.3 in Appendix A.1.

**Theorem 4.1** (Informal). *The regret of a policy extended to the augmented MDP is equal to the regret of the policy in the original MDP, and any policy defined in the augmented MDP can be converted into that in the original MDP without increasing the regret. Thus, solving the augmented MDP can yield correspondingly optimal policies for the original problem.*

**Remark 4.2.** *The benefit of using the equivalent $\overline{\mathcal{M}}$ is to avoid missing labels: given contexts in $D_{goal}$, the rewards in $\overline{\mathcal{M}}$ are known from our dataset setup in Section 3, whereas the rewards of the original MDP $\mathcal{M}$ are missing.*

## 4.2 Method

CODA is designed based on the observation on regret relationship in Theorem 4.1: As described in Figure 1, given a context-goal pair $(s, c)$ from the dataset $D_{\text{goal}}$, we create a fictitious transition from $s$ to $s^+$ with action $a^+$, reward $1$ under context $c$. We also label all unsupervised transitions in the dataset $D_{\text{dyn}}$ with the original action and reward $0$ under $c$. In this way, we can have a fully labeled transition dataset in the augmented MDP given any $c$ from the context-goal dataset and then run offline algorithms (based on the Bellman equation) on this dataset. This CODA algorithm is formally stated in Algorithm 1. It takes two datasets $D_{\text{dyn}}$ and $D_{\text{goal}}$ as input, and produces a labeled transition dataset $\bar{D}_{\text{dyn}} \bigcup \bar{D}_{\text{goal}}$ that is suitable for use by any offline RL algorithm based on Bellman equation like CQL [19], IQL [18], PSPI [37], ATAC [4], etc.

**Interpretation.** Why would our action augmentation make sense? We consider dynamic programming on the created dataset. Imagine we have a fictious transition from $s$ to $s^+$ with $a^+$ under context $c$. When we calculate $V^*(x)$ via Bellman equation where $x = (s, c)$, it will choose the action with the highest $Q^*$ value in the augmented action space. The fictitious action would be the optimal action since it induces the highest $Q^*$ value[2], meaning $s$ is already in $G_c$, and no further action is needed. Then the value of $V^*(x)$ would *naturally propagate to some state $x_{prev} = (s_{prev}, c)$ via Bellman equation if $x$ is reachable starting from $x_{prev}$* as shown in Figure 1, so $x_{prev}$ would still have meaningful values even with the intermediate reward $0$. For $x$ to be reachable starting from $x_{prev}$, we do not require the exact $s$ to appear in the trajectory dataset due to the generalization ability of the value function (details in Section 5). For non-goal states, such fictitious action never appears in the dataset, thus it would not be the optimal action in Bellman equation in pessimistic offline RL. For example, the fictitious action never appears as the candidate in argmax in algorithms like IQL, and would be punished as OOD actions in algorithms like CQL. We will prove this insight formally below in Section 5.

---

**Algorithm 1** CODA for CGO

**Input**: Dynamics dataset $D_{\text{dyn}}$, context-goal dataset $D_{\text{goal}}$

    **for** each sample $(s, c) \sim D_{\text{goal}}$ **do**

        Create transition[3]$(x, a^+, 1, x^+)$, where $x = (s, c)$ and $x^+ = (s^+, c)$, add it to $\bar{D}_{\text{goal}}$

    **end for**

    **for** each $(s, a, s') \sim D_{\text{dyn}}$ **do**

        **for** each $(\cdot, c) \sim \mathcal{D}_{\text{goal}}$ **do**

            Create transition $(x, a^+, 0, x')$, where $x = (s, c)$ and $x' = (s', c)$, add it to $\bar{D}_{\text{dyn}}$

        **end for**

    **end for**

**Output**: $\bar{D}_{\text{dyn}}$ and $\bar{D}_{\text{goal}}$

---

**Remark 4.3.** *We do not need to learn to perform $a^+$ for the policy in practice since it is only for fictitious transitions which is already inside the goal set in the original MDP. (From the proof of Lemma A.3, we know taking $a^+$ is always strictly worse than taking actions in the original action space $\mathcal{A}$.) Therefore, we simply use the original action space for policy modeling and only use the fictitious transitions in value learning. We note that in practice Algorithm 1 can be implemented as a pre-processing step in the minibatch sampling of a deep offline RL algorithm (as opposed to computing the full $\bar{D}_{dyn}$ and $\bar{D}_{goal}$ once before learning).*

---

[2]For all $a \neq a^+$ $Q^*$, $Q^*(x, a) < Q^*(x, a^+)$ when $\gamma < 1$. If $\gamma = 1$, the agent might also learn to travel to other goal states starting from $x$ with some probability, which is also acceptable in CGO.

## 5 CGO is Learnable with Positive Data Only

In Section 4, we show that a fully labeled dataset can be created in the augmented MDP without inducing extra approximation errors. But we still have no access to negative data, i.e., context and non-goal pairs. A natural question arises: *Can we learn to solve CGO problems with positive data only? What conditions are needed for CGO to be learnable with offline datasets?*

We show in theory that we do *not* need negative data to solve CGO problems by conducting a formal analysis for our method, instantiated with PSPI [37] as an example of the base algorithm. We present the detailed algorithm CODA+PSPI in Appendix A.3. This algorithm uses function classes $\mathcal{F} : \mathcal{S} \times \mathcal{A} \to \mathbb{R}$ and $\mathcal{G} : \mathcal{S} \to \mathbb{R}$ to model value functions and optimizes the policy given a policy class $\Pi$ based on absolute pessimism defined on initial states.

We present our assumptions and the main theoretical result as follows.

**Assumption 5.1** (Realizability). *We assume for any $\pi \in \Pi$, $Q^\pi \in \mathcal{F}$ and $R \in \mathcal{G}$, where $\mathcal{F}, \mathcal{G}$ are the function classes for action-value and reward respectively.*

**Assumption 5.2** (Completeness). *We assume: For any $f \in \mathcal{F}$, $g \in \mathcal{G}$ and $\pi \in \Pi$, $\max(g(x), f(x, \pi)) \in \mathcal{F}$; And for any $f \in \mathcal{F}$, $\pi \in \Pi$, $\mathcal{T}^\pi f(x, a) \in \mathcal{F}$, where $\mathcal{T}^\pi$ is a zero-reward Bellman backup operator with respect to $P(s'|s, a)$: $\mathcal{T}^\pi f(x, a) := \gamma \mathbb{E}_{x' \sim P(s'|s,a) \mathbb{1}(c'=c)}[f(x', \pi)]$.*

These two assumptions mean that the function classes $\mathcal{F}$ and $\mathcal{G}$ are expressive enough, which are standard assumptions in offline RL based on Bellman equation [37]. For deriving our main result, we define the coverage assumption needed below.

**Definition 5.3.** *We define the generalized concentrability coefficients:*

$$
\mathfrak{C}_{dyn}(\pi) := \max_{f, f' \in \mathcal{F}} \frac{\|f - \mathcal{T}^\pi f'\|^2_{\rho^\pi_{\notin G}}}{\|f - \mathcal{T}^\pi f'\|^2_{\mu_{dyn}}} \qquad and \qquad \mathfrak{C}_{goal}(\pi) := \max_{g \in \mathcal{G}} \frac{\|g - R\|^2_{\rho^\pi_{\in G}}}{\|g - R\|^2_{\mu_{goal}}} \tag{2}
$$

*where* $\|h\|^2_\mu := \mathbb{E}_{x \sim \mu}[h(x)^2]$, $\rho^\pi_{\notin G}(x, a) = \mathbb{E}_{\pi, P}\left[\sum_{t=0}^{T-1} \gamma^t \mathbb{1}(x_t = x, a_t = a)\right]$, $\rho^\pi_{\in G}(x) = \mathbb{E}_{\pi, P}\left[\gamma^T \mathbb{1}(x_T = x)\right]$, *and $T$ is the first time the agent enters the goal set.*

Concentrability coefficients is a generalization notion of density ratio: It describes how much the (unnormalized) distribution in the numerator is "covered" by that in the denominator in terms of the generalization ability of function approximators [37]. If $\mathfrak{C}_{dyn}(\pi), \mathfrak{C}_{goal}(\pi)$ are finite given $\mu_{goal}, \mu_{dyn}, \mathcal{F}, \mathcal{G}$ and $\pi$, then we say $\pi$ is covered by the data distributions, and conceptually offline RL can learn a policy to be no worse than $\pi$.

We now state our theoretical result, which is proven by a careful reformulation of the Bellman equation of the action-augmented MDP, and construct augmented value function and policy classes in the analysis using the CGO structures (see Appendix A).

**Theorem 5.4.** *Let $\pi^\dagger$ denote the learned policy of CODA + PSPI with datasets $D_{dyn}$ and $D_{goal}$, using value function classes $\mathcal{F} = \{\mathcal{X} \times \mathcal{A} \to [0, 1]\}$ and $\mathcal{G} = \{\mathcal{X} \to [0, 1]\}$. Under Assumption 5.1, 5.2 and 5.3, with probability $1 - \delta$, it holds, for any $\pi \in \Pi$,*

$$
J(\pi) - J(\pi^\dagger) \lesssim \mathfrak{C}_{dyn}(\pi) \left( \sqrt{\frac{\log(|\mathcal{F}||\mathcal{G}||\Pi|/\delta)}{|D_{dyn}|}} + \sqrt{\frac{\log(|\mathcal{F}||\mathcal{G}||\Pi|/\delta)}{|D_{goal}|}} \right) + \mathfrak{C}_{goal}(\pi) \sqrt{\frac{\log(|\mathcal{G}|/\delta)}{|D_{goal}|}}
$$

*where $\mathfrak{C}_{dyn}(\pi)$ and $\mathfrak{C}_{goal}(\pi)$ are concentrability coefficients[4].*

**Interpretation.** We can interpret Theorem 5.4 as follows: The statistical errors in value function estimation would decrease as we have more data from $\mu_{goal}$ and $\mu_{dyn}$; For any comparator $\pi$ with finite coefficients $\mathfrak{C}_{dyn}(\pi), \mathfrak{C}_{goal}(\pi)$, the final regret upper bound would also decrease. Taking $\pi = \pi^*$ as an example. For the coefficients $\mathfrak{C}_{dyn}(\pi), \mathfrak{C}_{goal}(\pi)$ to be finite, it indicates 1) the state-action distribution from the dynamics data "covers" the trajectories generated by $\pi^*$, which includes the case of stitching[5]; 2) the support of $\mu_{goal}$ "covers" the goals $\pi^*$ would reach. We note that these conditions are *not* any stronger than general requirements to solve offline algorithms: The "coverage"

---

[4]We state a more general result for non-finite function classes in Theorem A.11 in the appendix

[5]This does not mean the dynamics data have to be generated by the optimal policy; they can be generated by highly suboptimal policies so long as they together provide sufficient coverage.

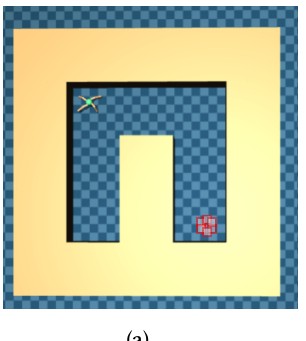
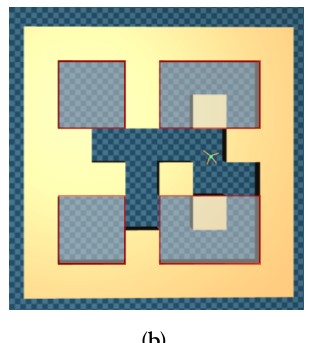
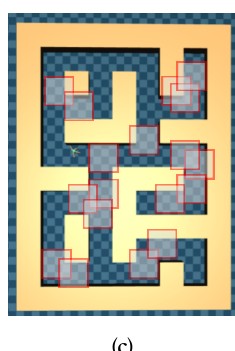

|(a)|(b)|(c)|

Figure 2: Illustration of the context-goal relationship with increasing complexity (Each red boundary defines a goal set with its center location as context). (a) Contexts and goal sets are very similar such that it could be approximately solved by a context-agnostic policy. (b) Contexts are finite, and different contexts map to distinct goal sets, which requires context-dependent policies. (c) Contexts are continuous and infinite. The context-goal mapping is neither one-to-many nor many-to-one, creating a CGO problem with full complexity.

above is measured based on the generalization ability of $f$ and $g$ respectively as in Definition 5.3; e.g., if $f(x_1)$ and $f(x_2)$ are similar for $x_1 \neq x_2$, then $x_2$ is within the coverage of $\mu$ so long as $x_1$ can be generated by $\mu$ in terms of the generalization ability of $f$. Such a coverage condition is weaker than coverage conditions based on density ratios. Besides, Theorem 5.4 simultaneously apply to all $\pi \in \Pi$ not just $\pi^*$. Therefore, as long as the above "coverage" conditions hold for any policy $\pi$ that can reach the goal set, the agent can learn to reach the goal set. Thus, we show that CODA with PSPI can provably solve CGO without the need for additional non-goal samples, i.e., CGO is learnable with positive data only.

**Remark 5.5.** *Here we only require function approximation assumptions made in the original MDP, without relying on functions defined on the fictitious action or completeness assumptions based on the fictitious transition. As a result, our theoretical results are comparable with those of other approaches.*

**Remark 5.6.** *MAHALO [21] is a SOTA offline RL algorithm that can provably learn from unlabeled data. One version of MAHALO is realized on top of PSPI in theory; however, their theoretical result (Theorem D.1) requires a stronger version concentrability, $\max_{g \in \mathcal{G}} \|g-r\|^2_{\rho^\pi_{\notin G}}/\|g-r\|^2_{\mu_{goal}}$, to be small. In other words, it needs negative examples of (context, non-goal state) tuples for learning.*

**Intuition for other base algorithms.**    Notice that PSPI is just one instantiation. Conceptually, the coverage conditions above also make sense for other pessimistic offline RL instantiations based on the Bellman equation (like IQL), since the key ideas used in the above analyses are that the regret relationship (Theorem 4.1) between the original MDP and the action augmented MDP (which is algorithm agnostic) and that pessimism together with Bellman equations can effectively propagate information from the context-goal dataset (without the need for negative data). However, performing complete theoretical analyses of CODA for all different offline RL algorithms is out of the scope of this paper.

## 6   Experiments

In this section, we present the experimental setup and results for CODA. Code is publicly available at: `https://github.com/yingfan-bot/coda`.

For a comprehensive empirical study, we first introduce the diverse spectrum of practical CGO setups.

**Diverse spectrum of practical CGO problems.** The main challenge of the CGO problem compared with traditional goal-conditioned RL is the potential complexity in the context-goal relationship. Therefore, to showcase the efficacy of different methods, we construct three levels with *increasing difficulty* as shown in Figure 2: (a) has a similar complexity as a single-task problem where the context does not play a significant role; (b) requires a context-dependent policy but only has finite contexts; (c) has infinite continuous context, requiring a context-dependent policy and generalization ability to contexts outside the offline data set. We aim to answer the following questions: 1) Does our method work under the data assumptions in Section 3, with different levels of context-goal complexity? 2) Is there any empirical benefit from using CODA, compared with baseline methods including reward learning, goal prediction, etc?

### 6.1 Environments and Datasets

**Dynamics dataset.** For all experiments, we use the original AntMaze-v2 datasets (3 different mazes and 6 offline datasets) of D4RL [6] as dynamics datasets $D_{\text{dyn}}$, removing all rewards and terminals.

**Context-goal dataset.** We construct three levels of context and goal relationships as shown in Figure 2. For each setup, we first define the context set, and then sample a fixed set of states from the offline trajectory dataset that satisfies the context-goal relationship, and then randomly *perturb* the states such that there would be no way to directly match goal examples to some states in the trajectories given contexts. Notice that this context-goal relationship is only used for dataset construction and is not accessible to the learning algorithm.[6] The specific context-goal relationship are discussed in Section 6.3 with the construction/evaluation details in Appendix B.2.

### 6.2 Method and Baselines

For controlled experiments, we use IQL [18] as the same backbone offline algorithm for all the methods with the same set of hyperparameters. Our choice of IQL is motivated by both its benchmarked performance on several RL domains and its structural similarity to PSPI (use of value/policy function classes along with pessimism). Please see Appendix B.1 for hyperparameters.

We describe the algorithms compared in the experiments.

**CODA.** We apply CODA in Algorithm 1 with IQL as the offline RL algorithm to solve the augmented MDP defined in Section 4.1 More specifically, we set $a^+$ to be an extra dimension in the action space of the action-value function, and model the policy with the original action space. Empirically, we found that equally balancing the samples $\bar{D}_{\text{dyn}}$ and $\bar{D}_{\text{goal}}$ generates the best result[7]. Then we apply IQL on this labeled dataset.

**Reward prediction.** For this family of baselines, we need to use the learned reward to predict the label of context-goal samples in the randomly sampled context-transition pairs during training, so we need to pre-train a reward model using the context-goal dataset. We use PDS [14] for reward modeling, and learn a *pessimistic* reward function using ensembles of models on the context-goal dataset. Then we apply the reward model to label the transitions with contexts, run IQL on this labeled dataset, and get a context-dependent policy. Besides PDS, we also test naive reward prediction (RP, which follows the same setup of PDS but without ensembles) and UDS [42] +RP in Section 6.3 (See details in Appendix B.1). Additionally, we add results from training with the oracle reward (marked as "Oracle Reward") where we provide the oracle reward for any query context-goal pairs, as a reference of the performance upper bound for reward prediction methods.

**Goal prediction.** We consider another GCRL-based baseline. Notice that the relationship between contexts and goals is unknown in CGO, we cannot directly apply traditional GCRL methods to CGO problems. Therefore, we adopt a workaround to use GCRL methods: We learn a conditional generative model as the goal predictor using classifier-free diffusion guidance [13], where the contexts serve as the condition, and the goal examples are used to train the generative model. We also learn a general goal-conditioned policy with the dynamics-only dataset using HER [1]+IQL. Given a test context, the goal predictor samples the goal given the context, which is then passed as the condition to the policy.

### 6.3 Results

**Original AntMaze: Figure 2(a).** In the original AntMaze, 2D goal locations (contexts) are limited to a small area as in Figure 2 (a). To make it a CGO problem, we make the test context visible to the agent. This setting in Figure 2 is approximately a single-task problem.

CODA generally achieves better performance than reward learning and goal prediction methods. Comparing the normalized return in each AntMaze environment for all methods, our method consistently achieves equivalent or better performance in each environment compared to other baselines

---

[6]Also note that the state space in Antmaze not only includes the 2D location; it also includes data from robotic arms, etc. We define the context-goal relationship only on the 2D location and ignore other information.

[7]We study the effect of this sampling ratio on CODA's performance in Table 5 in Appendix B.1

(Table 1). [8] Moreover, the performance of Goal Prediction is rather poor, which mainly comes from not enough goal examples to learn from in this setup due to a limited goal area.

Table 1: Average success rate (%) in AntMaze-v2 from all environments.

| Env/Method | CODA (Ours) | PDS | Goal Prediction | RP | UDS+RP | Oracle Reward |
|---|---|---|---|---|---|---|
| umaze | **94.8±1.3** | 93.0±1.3 | 46.4±6.0 | 50.5±2.1 | 54.3±6.3 | 94.4±0.61 |
| umaze diverse | **72.8±7.7** | 50.6±7.8 | 42.8±4.4 | **72.8±2.6** | 71.5±4.3 | 76.8±5.44 |
| medium play | **75.8±1.9** | 66.8±4.9 | 43.8±4.7 | 0.5±0.3 | 0.3±0.3 | 80.6±1.56 |
| medium diverse | **84.5±5.2** | 22.8±2.4 | 28.6±3.9 | 0.5±0.5 | 0.8±0.5 | 72.4±4.26 |
| large play | **60.0±7.6** | 39.6±4.9 | 13.0±4.0 | 0±0 | 0±0 | 41.2±3.58 |
| large diverse | **36.8±6.9** | 30.0±5.3 | 12.6±2.7 | 0±0 | 0±0 | 34.2±2.59 |
| average | **70.8** | 50.5 | 31.2 | 20.7 | 21.2 | 66.6 |

**Four Rooms: Figure 2(b).** We partition the maze into four rooms as in Figure 2(b), where the discrete room numbers (1,2,3,4) serve as contexts and we uniformly select test contexts. A context-dependent policy is needed, but there is no generalization required for unseen contexts in this setup.

We show the normalized return (average success rate in percentage) in each modified Four Rooms environment for our method and baseline methods in Table 2, where our method consistently outperforms the performances of baseline methods.

Table 2: Average scores from Four Rooms with perturbation. The score for each run is the average success rate (%) of the other three rooms.

| Env/Method | CODA (Ours) | PDS | Goal Prediction | Oracle Reward |
|---|---|---|---|---|
| medium-play | **78.7±0.9** | 46.0±4.47 | 59.3±2.6 | 77.7±2.0 |
| medium-diverse | **83.6±1.9** | 51.3±3.6 | 66.7±2.4 | 87.4±1.2 |
| large-play | **65.5±2.5** | 13.9±2.4 | 41.4±3.6 | 67.2±2.7 |
| large-diverse | **72.2±2.9** | 11.1±3.8 | 42.0±3.0 | 69.6±3.1 |
| average | **75.0** | 30.6 | 52.4 | 75.5 |

**Random Cells: Figure 2(c).** We use a diverse distribution of contexts as shown in Figure 2(c), where the contexts are randomly sampled from non-wall states. For test contexts, we have two settings: 1) sampling from the training distribution; 2) sampling from a far-away area from the start states.

Overall, CODA outperforms the baselines under the setup in Figure 2(c). We show the normalized return (average success rate in percentage) in each modified Random Cells environment in Table 3, which also shows the generalization ability of our method in the context space. CODA also generalizes to a different test context distribution: We also test with a distribution shift of the contexts in Table 4. We can observe that when tested with this different context distribution, CODA still generates better overall results compared to reward learning and goal prediction baselines.

Table 3: Average scores from Random Cells. The score for each run is the average success rate (%) of random test contexts from the same training distribution.

| Env/Method | CODA (Ours) | PDS | Goal Prediction | Oracle Reward |
|---|---|---|---|---|
| medium-play | **76.8±6.1** | 52.0±8.8 | 66.7±7.2 | 71.9±0.1 |
| medium-diverse | **78.2±6.5** | 60.9±11.3 | 69.7±8.7 | 79.3±6.1 |
| large-play | **57.6±12.4** | 50.6±6.4 | 42.4±8.2 | 49.4±9.3 |
| large-diverse | 54.7±8.8 | **58.3±9.2** | 44.2±8.1 | 58.2±3.4 |
| average | **66.8** | 55.5 | 55.8 | 64.7 |

**Reference to training with oracle reward.** Notice that training with oracle reward is the skyline performance. From the results, training with oracle reward does not generally improve the performance much compared to CODA, though it generally outperforms PDS and Goal Prediction. This is mainly due to the sparsity of the positive samples in the randomly sampled context-transition pairs. On the other hand, CODA easily uses these positive examples via our augmentation, which is another advantage of our method over reward prediction baselines.

---

[8]We find umaze is too easy: even if the reward labeling is bad it still has a relatively high reward, so we also omit it in other experiments. We also find UDS and RP are not very effective in our data setup, so we also omit them in other experiments.

Table 4: Average scores from Random Cells with perturbation. The score for each run is the average success rate (%) of random test contexts with a distribution shift.

| Env/Method | CODA (Ours) | PDS | Goal Prediction | Oracle Reward |
|---|---|---|---|---|
| medium-play | 67.9±8.2 | 50.1±13.4 | **70.5±1.9** | 67.2±7.2 |
| medium-diverse | **72.5±6.5** | 57.5±14.8 | 63.0±7.2 | 68.7±7.9 |
| large-play | **60.2±4.8** | 48.1±8.0 | 44.3±4.1 | 59.8±4.4 |
| large-diverse | **58.0±5.8** | 44.1±9.9 | 55.4±5.7 | 57.6±7.6 |
| average | **64.7** | 49.9 | 58.3 | 63.3 |

**Evaluation of the Reward Model.** We also visualize the learned reward model from reward learning baselines in Appendix B.3: PDS is consistently better at separating positive and negative datasets than UDS and naive RP, but PDS can still fail at fully separating positive and negative examples. Intuitively, our method does not require reward learning thanks to the construction of the augmented MDP, which avoids the extra errors in reward prediction and leads to better performance.

### 6.4 Discussion and Limitation

Our experiments are limited to low-dimensional simulations. Nevertheless, the success of our method with diverse context-goal relationships serves as a first milestone to showcase its effectiveness, and we believe CODA would be useful in real-world settings (e.g., learning visual-language robot policies) for its simplicity and theoretical guarantees. Potential scaling up by incorporating features from large pretrained models would be an exciting future direction, which can make our method generalizable to the real world.

## 7 Conclusion

We propose CODA for offline CGO problems, and prove CODA can learn near-optimal policies without the need for negative labels with natural assumptions. We also validate the efficacy of CODA experimentally, and find it outperforms other reward-learning and goal prediction baselines across various CGO complexities. We believe our method has the potential to generalize to real-world applications by further scaling up.

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

# A  Theoretical Analysis

In this section, we provide a detailed analysis for the instantiation of CODA using PSPI [37]. We follow the same notation for the value functions, augmented MDP, and extended function classes as stated in Section 3 and Section 4 in the main text.

## A.1  Equivalence Relations between Original and Augmented MDP

We begin by showing that the optimal policy and any value function in the augmented MDP can be expressed using their analog in the original MDP. With the augmented MDP defined as $\overline{\mathcal{M}} := (\bar{\mathcal{X}}, \bar{\mathcal{A}}, \bar{R}, \bar{P}, \gamma)$ in Section 4.1, we first define the value function in the augmented MDP. For a policy $\bar{\pi} : \bar{\mathcal{X}} \to \bar{\mathcal{A}}$, we define the Q function for the augmented MDP as

$$\bar{Q}^{\bar{\pi}}(x, a) := \mathbb{E}_{\bar{\pi}, \bar{P}} \left[ \sum_{t=0}^{\infty} \gamma^t \bar{R}(x, a) | x_0 = x, a_0 = a \right]$$

Notice that we don't have a reaching time random variable $T$ in this definition; instead the agent would enter an absorbing state $s^+$ after taking $a^+$ in the augmented MDP. We can define similarly $\bar{V}^{\bar{\pi}}(s) := \bar{Q}^{\bar{\pi}}(x, \bar{\pi})$.

**Remark A.1.** *Let $\bar{Q}_R^{\pi}$ be the extension of $Q^{\pi}$ based on $R$. We have, for $x \notin G$, $\bar{Q}_R^{\pi}(x, a) = \bar{Q}^{\bar{\pi}}(x, a)$ $\forall a \in \bar{\mathcal{A}}$, and for $x \in G$, $\bar{Q}_R^{\pi}(x, a) = \bar{Q}^{\bar{\pi}}(x, a^+) = 1$, $\forall a \in \bar{\mathcal{A}}$.*

By the construction of the augmented MDP, it is obvious that the following is true.

**Lemma A.2.** *Given $\pi : \mathcal{X} \to \Delta(\mathcal{A})$, let $\bar{\pi}$ be its extension. For any $h : \mathcal{X} \times \mathcal{A} \to \mathbb{R}$, it holds*

$$\mathbb{E}_{\pi, P} \left[ \sum_{t=0}^{T} \gamma^t h(x, a) \right] = \mathbb{E}_{\bar{\pi}, \bar{P}} \left[ \sum_{t=0}^{\infty} \gamma^t \tilde{h}^{\pi}(x, a) | x \notin \mathcal{X}^+ \right]$$

*where $T$ is the goal-reaching time (random variable) and we define $\tilde{h}^{\pi}(x, a^+) = h(x, \pi)$.*

We can now relate the value functions between the two MDPs.

**Proposition A.3.** *For a policy $\pi : \mathcal{X} \to \Delta(\mathcal{A})$, let $\bar{\pi}$ be its extension (defined above). We have for all $x \in \mathcal{X}$, $a \in \mathcal{A}$,*

$$Q^{\pi}(x, a) \geq \bar{Q}^{\bar{\pi}}(x, a)$$
$$V^{\pi}(x) = \bar{V}^{\bar{\pi}}(x)$$

*Conversely, for a policy $\xi : \bar{\mathcal{X}} \to \Delta(\bar{\mathcal{A}})$, define its restriction $\underline{\xi}$ on $\mathcal{X}$ and $\mathcal{A}$ by translating probability of $\xi$ originally on $a^+$ to be uniform over $\mathcal{A}$. Then we have for all $s \in \mathcal{S}$, $a \in \mathcal{A}$*

$$Q^{\underline{\xi}}(x, a) \geq \bar{Q}^{\xi}(x, a)$$
$$V^{\underline{\xi}}(x) \geq \bar{V}^{\xi}(x)$$

*Proof.* The first direction follows from Lemma A.2. For the latter, whenever $\xi$ takes $a^+$ at some $x \notin G$, it has $\bar{V}^{\xi}(x) = 0$ but $\bar{V}^{\underline{\xi}}(x) \geq 0$ since there is no negative reward in the original MDP. By performing a telescoping argument, we can derive the second claim. $\square$

By this lemma, we know the extension of $\pi^*$ (i.e., $\bar{\pi}^*$) is also optimal to the augmented MDP and $V^*(x) = \bar{V}^*(x)$ for $x \in \mathcal{X}$. Furthermore, we have a reduction that we can solve for the optimal policy in the original MDP by the solving augmented MDP, since

$$V^{\underline{\xi}}(d_0) - V^*(d_0) \leq V^{\xi}(d_0) - \bar{V}^*(d_0)$$

for all $\xi : \bar{\mathcal{X}} \to \Delta(\bar{\mathcal{A}})$. In particular,

$$\text{Regret}(\pi) := V^{\pi}(d_0) - V^*(d_0) = V^{\bar{\pi}}(d_0) - \bar{V}^*(d_0) =: \overline{\text{Regret}}(\bar{\pi}) \tag{3}$$

Since the augmented MDP replaces the random reaching time construction with an absorbing-state version, the Q function $\bar{Q}^{\bar{\pi}}$ of the extended policy $\bar{\pi}$ satisfies the Bellman equation

$$\bar{Q}^{\bar{\pi}}(x, a) = \bar{R}(x, a) + \gamma \mathbb{E}_{x' \sim \bar{P}(\cdot|x,a)}[\bar{Q}^{\bar{\pi}}(x', \bar{\pi})]$$
$$=: \bar{\mathcal{T}}^{\bar{\pi}} \bar{Q}^{\bar{\pi}}(x, a) \tag{4}$$

For $x \in \mathcal{X}$ and $a \in \mathcal{A}$, we show how the above equation can be rewritten in $Q^{\pi}$ and $R$.

**Proposition A.4.** *For $x \in \mathcal{X}$ and $a \in \mathcal{A}$,*

$$\bar{Q}^{\bar{\pi}}(x, a) = 0 + \gamma \mathbb{E}_{x' \sim \bar{P}(\cdot|x,a)}[\max(R(x'), Q^{\pi}(x', \pi))]$$

*For $a = a^+$, $\bar{Q}^{\bar{\pi}}(x, a^+) = \bar{R}(x, a^+) = R(x)$. For $x \in \mathcal{X}^+$, $\bar{Q}^{\bar{\pi}}(x, a) = 0$.*

*Proof.* The proof follows from Lemma A.5 and the definition of $\bar{P}$. □

**Lemma A.5.** *For $x \in \mathcal{X}$, $\bar{Q}^{\bar{\pi}}(x, \bar{\pi}) = \max(R(x), Q^{\pi}(x, \pi))$*

*Proof.* For $x \in \mathcal{X}$,

$$\bar{Q}^{\bar{\pi}}(x, \bar{\pi}) = \begin{cases} \bar{Q}^{\bar{\pi}}(x, a^+), & \text{if } x \in G \\ \bar{Q}^{\bar{\pi}}(x, \pi), & \text{otherwise} \end{cases} \qquad \text{(Because of definition of } \bar{\pi})$$

$$= \begin{cases} \bar{Q}^{\bar{\pi}}(x, a^+), & \text{if } x \in G \\ Q^{\pi}(x, \pi), & \text{otherwise} \end{cases} \qquad \text{(Because of Proposition A.3)}$$

$$= \begin{cases} \bar{R}(x, a^+), & \text{if } x \in G \\ Q^{\pi}(x, \pi), & \text{otherwise} \end{cases} \qquad \text{(Definition of augmented MDP)}$$

$$= \begin{cases} R(x), & \text{if } x \in G \\ Q^{\pi}(x, \pi), & \text{otherwise} \end{cases}$$

$$= \max(R(x), Q^{\pi}(x, \pi))$$

where in the last step we use $\bar{R}(x) = 1$ for $x \in G$ and $\bar{R}(x) = 0$ otherwise. □

## A.2 Function Approximator Assumptions

In Theorem 5.4, we assume access to a policy class $\Pi = \{\pi : \mathcal{X} \to \Delta(\mathcal{A})\}$. We also assume access to a function class $\mathcal{F} = \{f : \mathcal{X} \times \mathcal{A} \to [0, 1]\}$ and a function class $\mathcal{G} = \{g : \mathcal{X} \to [0, 1]\}$. We can think of them as approximators for the Q function and the reward function of the original MDP.

For an action value function $f : \mathcal{X} \times \mathcal{A} \to [0, 1]$, define its extension:

$$\bar{f}_g(x, a) = \begin{cases} g(x), & a = a^+ \text{ and } x \notin \mathcal{X}^+ \\ 0, & x \in \mathcal{X}^+ \\ f(x, a), & \text{otherwise.} \end{cases} \tag{5}$$

The extension of $f$ is based on a state value function $g : \mathcal{X} \to [0, 1]$ which determines the action value of $x$ only at $a^+$. One could also view $g(x)$ as a goal indicator: after taking $a^+$ the agent would always transit to the zero-reward absorbing state $s^+$, so $g(x) = \bar{R}(x, a^+)$ which is the indicator of whether $s \in G_c$.

Recall the zero-reward Bellman backup operator $\mathcal{T}^{\pi}$ with respect to $P(s'|s, a)$ as defined in Assumption 5.2:

$$\mathcal{T}^{\pi} f(x, a) := \gamma \mathbb{E}_{x' \sim P_0(\cdot|x,a)}[f(x', \pi)]$$

where $P_0(x'|x, a) := P(s'|s, a)\mathbb{1}(c' = c)$. Note this definition is different from the one with absorbing state $s^+$ in Section 3. Using this modified backup operator, we can show that the following realizability assumption is true for the augmented MDP:

**Proposition A.6** (Realizability). *By Assumption 5.1 and Assumption 5.2, there is $f \in \mathcal{F}$ and $g \in \mathcal{G}$ such that $\bar{Q}^{\bar{\pi}} = \bar{f}_g$.*

*Proof.* By Assumption 5.2, there is $h \in \mathcal{F}$ such that $h(x,a) = \max(R(x), Q^\pi(x,a))$. By Proposition A.4, we have for $x \in \mathcal{X}$, $a \neq a^+$

$$\bar{Q}^{\bar{\pi}}(x,a) = 0 + \gamma \mathbb{E}_{x' \sim \bar{P}(\cdot|x,a)}[\max(R(x'), Q^\pi(x',\pi))]$$
$$= 0 + \gamma \mathbb{E}_{x' \sim P_0(\cdot|x,a)}[h(x,\pi)]$$
$$= \mathcal{T}^\pi h \in \mathcal{F}$$

For $a = a^*$, we have $\bar{Q}^{\bar{\pi}}(x,a^*) = \bar{R}(x,a^+) = R(x) \in \mathcal{G}$. Finally $\bar{Q}^{\bar{\pi}}(x^+,a) = 0$ for $x^+ \in \mathcal{X}^+$. Therefore, $\bar{Q}^{\bar{\pi}} = \bar{f}_g$ for some $f \in \mathcal{F}$ and $g \in \mathcal{G}$. $\qquad\square$

### A.3 CODA+PSPI Algorithm

In this section, we describe the instantiation of PSPI with CODA in detail along with the necessary notation. The main theoretical result and its proof is then given in Section A.4. As discussed in Section 5, our algorithm is based on the idea of reduction, which turns the offline CGO problem into a standard offline RL problem in the augmented MDP. To this end, we construct augmented datasets $\bar{D}_{\text{dyn}}$ and $\bar{D}_{\text{goal}}$ in Algorithm 1 as follows:

$$\bar{D}_{\text{dyn}} = \{(x_n, a_n, r_n, x_n') | r_n = 0, x_n = (s_i, c_j), x_n' = (s_i', c_j), a_n = a_i, (s_i, a_i, s_i') \in D_{\text{dyn}}, (\cdot, c_j) \in D_{\text{goal}}\}$$
$$\bar{D}_{\text{goal}} = \{(x_n, a^+, r_n, x_n^+) | r_n = 1, x_n = (s_n, c_n), x_n^+ = (s^+, c_n), (s_n, c_n) \in D_{\text{goal}}\}$$

With this construction, we have: $\bar{D}_{\text{dyn}} \sim \mu_{\text{dyn}}(s,a,s')\mu_{\text{goal}}(c)$ and $\bar{D}_{\text{goal}} \sim \mu_{\text{goal}}(c,s)\mathbb{1}(a = a^+)\mathbb{1}(s' = s^+)$. We use the notation, $\bar{\mu}_{\text{dyn}}(x,a,x') = \mu_{\text{dyn}}(s,a,s')\mu_{\text{goal}}(c)$ and $\bar{\mu}_{\text{goal}}(x,a,x') = \mu_{\text{goal}}(c,s)\mathbb{1}(a = a^+)\mathbb{1}(s' = s^+)$. We will also use the notation $x_{ij} \equiv (s_i, c_j)$, $x_{ij}' \equiv (s_i', c_j)$ in the above construction. These two datasets have the standard tuple format, so we can run offline RL on $\bar{D}_{\text{dyn}} \bigcup \bar{D}_{\text{goal}}$. Also, note that $|\bar{D}_{\text{dyn}}| = |D_{\text{dyn}}||D_{\text{goal}}|$ and $|\bar{D}_{\text{goal}}| = |D_{\text{goal}}|$.

**PSPI.** We consider the information theoretic version of PSPI [37] which can be summarized as follows: For an MDP $(\mathcal{X}, \mathcal{A}, R, P, \gamma)$, given a tuple dataset $D = \{(x,a,r,x')\}$, a policy class $\Pi$, and a value class $\mathcal{F}$, it finds the policy through solving the two-player game:

$$\max_{\pi \in \Pi} \min_{f \in \mathcal{F}} \quad f(d_0, \pi) \qquad \text{s.t.} \qquad \ell(f,f;\pi,D) - \min_{f' \in \mathcal{F}} \ell(f',f;\pi,D) \leq \epsilon_b \qquad (6)$$

where $f(d_0,\pi) = \mathbb{E}_{x_0 \sim d_0}[f(x_0,\pi)]$, $\ell(f,f';\pi,D) := \frac{1}{|D|}\sum_{(x,a,r,x') \in D}(f(x,a) - r - f'(x',\pi))^2$. The term $\ell(f,f;\pi,D) - \min_{f'} \ell(f',f;\pi,D)$ in the constraint is an empirical estimation of the Bellman error on $f$ with respect to $\pi$ on the data distribution $\mu$, i.e. $\mathbb{E}_{x,a \sim \mu}[(f(x,a) - \mathcal{T}^\pi f(x,a))^2]$. It constrains the Bellman error to be small, since $\mathbb{E}_{x,a \sim \mu}[(Q^\pi(x,a) - \mathcal{T}^\pi Q^\pi(x,a))^2] = 0$.

**CODA+PSPI.** Below we show how to run PSPI to solve the augmented MDP with offline dataset $\bar{D}_{\text{dyn}} \bigcup \bar{D}_{\text{goal}}$. To this end, we extend the policy class from $\Pi$ to $\bar{\Pi}$, and the value class from $\mathcal{F}$ to $\bar{\mathcal{F}}_\mathcal{G}$ using the function class $\mathcal{G}$ based on the extensions defined in Section 4.1. One natural attempt is to implement equation 6 with the extended policy and value classes $\bar{\Pi}$ and $\bar{\mathcal{F}}$ and $\bar{D} = \bar{D}_{\text{dyn}} \bigcup \bar{D}_{\text{goal}}$. This would lead to the two player game:

$$\max_{\bar{\pi} \in \bar{\Pi}} \min_{\bar{f}_g \in \bar{\mathcal{F}}_\mathcal{G}} \quad \bar{f}_g(d_0, \bar{\pi}) \qquad \text{s.t.} \qquad \ell(\bar{f}_g, \bar{f}_g; \bar{\pi}, \bar{D}) - \min_{\bar{f}_{g'}' \in \bar{\mathcal{F}}_\mathcal{G}} \ell(\bar{f}_{g'}', \bar{f}_g; \bar{\pi}, \bar{D}) \leq \epsilon_b \qquad (7)$$

However, equation 7 is not a well-defined algorithm, because its usage of the extended policy $\bar{\pi}$ in the constraint requires knowledge of $G$, which is unknown to the agent.

Fortunately, we show that equation 7 can be slightly modified so that the implementation does not actually require knowing $G$. Here we use a property (Proposition A.4) that the Bellman equation of the augmented MDP:

$$\bar{Q}^{\bar{\pi}}(x,a) = \bar{R}(x,a) + \gamma \mathbb{E}_{x' \sim \bar{P}(\cdot|x,a)}[\bar{Q}^\pi(x',\bar{\pi})]$$
$$= 0 + \gamma \mathbb{E}_{x' \sim \bar{P}(\cdot|x,a)}[\max(R(x'), Q^\pi(x',\pi))]$$

for $x \in \mathcal{X}$ and $a \neq a^+$, and $\bar{Q}^{\bar{\pi}}(x,a) = 1$ for $x \in G$ and $a = a^+$.

We can rewrite the squared Bellman error on these two data distributions, $\bar{D}_{\text{dyn}}$ and $\bar{D}_{\text{goal}}$, using the Bellman backup defined on the augmented MDP (see eq.4) as below:

$$\mathbb{E}_{\mu_{\text{dyn}}}[(\bar{Q}^{\bar{\pi}}(x,a) - \bar{\mathcal{T}}^{\bar{\pi}}\bar{Q}^{\bar{\pi}}(x,a))^2] = \mathbb{E}_{\mu_{\text{dyn}}}[(\bar{Q}^{\bar{\pi}}(x,a) - 0 - \gamma \mathbb{E}_{x' \sim \bar{P}(\cdot|x,a)}[\max(R(x), Q^\pi(x,\pi))])^2]$$

$$\mathbb{E}_{x,a \sim \mu_{\text{goal}}}[(\bar{Q}^{\bar{\pi}}(x,a) - \bar{\mathcal{T}}^{\bar{\pi}} \bar{Q}^{\bar{\pi}}(x,a))^2] = \mathbb{E}_{x,a \sim \mu_{\text{goal}}}[(\bar{Q}^{\bar{\pi}}(x,a^+) - 1)^2]$$

We can construct an approximator $\bar{f}_g(x,a)$ for $\bar{Q}^{\bar{\pi}}(x,a)$. Substituting the estimator $\bar{f}_g(x,a)$ for $\bar{Q}^{\bar{\pi}}(x,a)$ in the squared Bellman errors above and approximating them by finite samples, we derive the empirical losses below.

$$\ell_{\text{dyn}}(\bar{f}_g, \bar{f}'_{g'}; \bar{\pi}) := \frac{1}{|\bar{D}_{\text{dyn}}|} \sum_{(x,a,r,x') \in \bar{D}_{\text{dyn}}} (f(x,a) - \gamma \max(g'(x'), f'(x',\pi)))^2 \tag{8}$$

$$\ell_{\text{goal}}(\bar{f}_g) := \frac{1}{|\bar{D}_{\text{goal}}|} \sum_{(x,a,r,x') \in \bar{D}_{\text{goal}}} (g(x) - 1)^2 \tag{9}$$

where we have $\bar{f}_g(x,a) = f(x,a)\mathbb{1}(a \neq a^+) + g(x)\mathbb{1}(a = a^+)$ for $x \notin \mathcal{X}^+$.

Using this loss, we define the two-player game of PSPI for the augmented MDP:

$$\max_{\pi \in \Pi} \min_{\bar{f}_g \in \bar{\mathcal{F}}} \bar{f}_g(d_0, \bar{\pi}) \tag{10}$$

$$\text{s.t.} \quad \ell_{\text{dyn}}(\bar{f}_g, \bar{f}_g; \bar{\pi}) - \min_{\bar{f}'_{g'} \in \bar{\mathcal{F}}} \ell_{\text{dyn}}(\bar{f}'_{g'}, \bar{f}_g; \bar{\pi}) \leq \epsilon_{\text{dyn}}$$

$$\ell_{\text{goal}}(\bar{f}_g) \leq 0$$

Notice $\bar{f}_g(d_0, \bar{\pi}) = f(d_0, \pi)$. Therefore, this problem can be solved using samples from $D$ without knowing $G$.

## A.4 Analysis of CODA+PSPI

**Covering number.** We first define the covering number on the function classes $\mathcal{F}$, $\mathcal{G}$, and $\Pi$[9]. For $\mathcal{F}$ and $\mathcal{G}$, we use the $L_\infty$ metric. We use $\mathcal{N}_\infty(\mathcal{F}, \epsilon)$ and $\mathcal{N}_\infty(\mathcal{G}, \epsilon)$ to denote the their $\epsilon$-covering numbers. For $\Pi$, we use the $L_\infty$-$L_1$ metric, i.e., $\|\pi_1 - \pi_2\|_{\infty,1} := \sup_{x \in \mathcal{X}} \|\pi_1(\cdot|s) - \pi_2(\cdot|s)\|_1$. We use $\mathcal{N}_{\infty,1}(\Pi, \epsilon)$ to denote its $\epsilon$-covering number.

**High-probability events.** In CODA+PSPI (eq. 10), we choose the policy in class $\pi$ which has the best *pessimistic* value function estimate. In order to show this, we will need two high probability results (we defer their proofs to Section A.4.1). To that end, we will use the following notation for the expected value of the empirical losses:

$$\ell_{\bar{\mu}_{\text{dyn}}}(\bar{f}_g, \bar{f}'_{g'}; \bar{\pi}) := \mathbb{E}_{(x,a,x') \sim \bar{\mu}_{\text{dyn}}}(f(x,a) - \gamma \max(g'(x'), f'(x',\pi)))^2$$

$$\ell_{\bar{\mu}_{\text{goal}}}(\bar{f}_g) := \mathbb{E}_{(x,a^+,x^+) \sim \bar{\mu}_{\text{goal}}}(g(x) - 1)^2$$

First, we show that for any policy $\pi \in \Pi$, the true value function $\bar{Q}^{\bar{\pi}}$ satisfies the two empirical constraints specified in eq. equation 10.

**Lemma A.7.** *With probability at least $1 - \delta$, it holds for all $\pi \in \Pi$,*

$$\ell_{dyn}(\bar{Q}^{\bar{\pi}}, \bar{Q}^{\bar{\pi}}; \bar{\pi}) - \min_{\bar{f}'_{g'} \in \bar{\mathcal{F}}} \ell_{dyn}(\bar{f}'_{g'}, \bar{Q}^{\bar{\pi}}; \bar{\pi}) \leq O\left(\left(\sqrt{\frac{\square}{|D_{dyn}|}} + \sqrt{\frac{\square}{|D_{goal}|}}\right)^2\right)$$

$$\ell_{goal}(\bar{Q}^{\bar{\pi}}) \leq 0$$

*where*[10] $\square \equiv \log\left(\frac{\mathcal{N}_\infty(\mathcal{F}, 1/|D_{goal}||D_{dyn}|)\mathcal{N}_\infty(\mathcal{G}, 1/|D_{goal}||D_{dyn}|)\mathcal{N}_{\infty,1}(\Pi, 1/|D_{goal}||D_{dyn}|)}{\delta}\right)$.

---

[9]For finite function classes, the resulting performance guarantee will depend on $|\mathcal{F}|$, $|\mathcal{G}|$ and $|\Pi|$ instead of the covering numbers as stated in Theorem 5.4.

[10]Technically, we can remove $\mathcal{N}_\infty\left(\mathcal{G}, \frac{1}{|D_{\text{dyn}}||D_{\text{goal}}|}\right)$ in the upper bound, but we include it here for a cleaner presentation.

We use the notation $\epsilon_{\text{dyn}} := \left( \sqrt{\frac{\square}{|D_{\text{dyn}}|}} + \sqrt{\frac{\square}{|D_{\text{goal}}|}} \right)^2$ for the first upper bound in Lemma A.7.

Next, we show that for every pair of value function $\bar{f}_g \in \bar{\mathcal{F}}$ and policy $\bar{\pi} \in \bar{\Pi}$ which satisfies the constraints in eq. equation 10, the empirical estimates provide a bound on the population error with high probability.

**Lemma A.8.** *For all $f \in \mathcal{F}, g \in \mathcal{G}$ and $\pi \in \Pi$ satisfying*

$$\ell_{dyn}(\bar{f}_g, \bar{f}_g; \bar{\pi}) - \min_{\bar{f}'_{g'} \in \bar{\mathcal{F}}} \ell_{dyn}(\bar{f}'_{g'}, \bar{f}_g; \bar{\pi}) \le \epsilon_{dyn}$$

$$\ell_{goal}(\bar{f}_g) \le 0,$$

*with probability at least $1 - \delta$, we have:*

$$\left\| \bar{f}_g(x, a) - \gamma \mathbb{E}_{x' \sim \bar{P}(\cdot | x, a)} \left[ \max(g(x'), f(x', \pi)) \right] \right\|_{\bar{\mu}_{dyn}} \le O\left( \sqrt{\epsilon_{dyn}} \right)$$

$$\|g(x) - 1\|_{\bar{\mu}_{goal}} \le O\left( \sqrt{\frac{\log \frac{\mathcal{N}_\infty(\mathcal{G}, 1/|D_{goal}|)}{\delta}}{|D_{goal}|}} \right) =: \sqrt{\epsilon_{goal}}$$

**Pessimistic estimate.** Our next step is to show that the solution of the constrained optimization problem in equation 10 is pessimistic and that the amount of pessimism is bounded.

**Lemma A.9.** *Given $\pi$, let $\bar{f}_g^\pi$ denote the minimizer in equation 10. With high probability, $\bar{f}_g^\pi(d_0, \bar{\pi}) \le Q^\pi(d_0, \pi)$*

*Proof.* By Lemma A.7, for any policy $\pi \in \Pi$, we know that $\bar{Q}_R^{\bar{\pi}}$ satisfies the constraints in equation equation 10. Therefore, we have

$$\bar{f}_g^\pi(d_0, \bar{\pi}) \le \bar{Q}_R^\pi(d_0, \bar{\pi}) = Q^\pi(d_0, \pi).$$

$\square$

We will now bound the amount of underestimation for the minimizer $\bar{f}_g^\pi$ in the above lemma.

**Lemma A.10.** *Suppose $x_0 \sim d_0$ is not in $G$ almost surely. For any $\pi \in \Pi$,*

$$Q^\pi(d_0, \pi) - \bar{f}_g^\pi(d_0, \bar{\pi})$$
$$\le \mathbb{E}_\pi \left[ \sum_{t=0}^{T-1} \gamma^t \left( \gamma \max(g^\pi(x_{t+1}), f^\pi(x_{t+1}, \pi)) - f^\pi(x_t, a_t) \right) + \gamma^T (R(x_T) - g^\pi(x_T)) \right]$$

*Note that in a trajectory $x_T \in G$ whereas $x_t \notin G$ for $t < T$ by definition of $T$.*

*Proof.* Let $\bar{f}_g^\pi = (f^\pi, g^\pi)$ be the empirical minimizer. By performance difference lemma, we can write

$$(1 - \gamma)Q^\pi(d_0, \pi) - (1 - \gamma)\bar{f}_g^\pi(d_0, \bar{\pi}) = (1 - \gamma)\bar{Q}^\pi(d_0, \bar{\pi}) - (1 - \gamma)\bar{f}_g^\pi(d_0, \bar{\pi})$$
$$= \mathbb{E}_{\bar{d}^{\bar{\pi}}}[\bar{R}(x, a) + \gamma \bar{f}_g^\pi(x', \bar{\pi}) - \bar{f}_g^\pi(x, a)]$$

where with abuse of notation we define $\bar{d}^{\bar{\pi}}(x, a, x') := \bar{d}^{\bar{\pi}}(x, a)\bar{P}(x'|x, a)$, where $\bar{d}^{\bar{\pi}}(x, a)$ is the average state-action distribution of $\bar{\pi}$ in the augmented MDP.

In the above expectation, for $x \in G$, we have $a = a^+$ and $x^+ = (s^+, c)$ after taking $a^+$ at $x = (s, c)$, which leads to

$$\bar{R}(x, a) + \gamma \bar{f}_g^\pi(x', \bar{\pi}) - \bar{f}_g^\pi(x, a) = \bar{R}(x, a^+) + \gamma \bar{f}_g^\pi(x^+, \bar{\pi}) - \bar{f}_g^\pi(x, a^+) = R(x) - g^\pi(x)$$

For $x \notin G$ and $x \notin \mathcal{X}^+$, we have $a \ne a^+$ and $x' \notin \mathcal{X}^+$; therefore

$$\bar{R}(x, a) + \gamma \bar{f}_g^\pi(x', \bar{\pi}) - \bar{f}_g^\pi(x, a) = R(x) + \gamma \bar{f}_g^\pi(x', \bar{\pi}) - f^\pi(x, a)$$
$$\le \gamma \max(g^\pi(x'), f^\pi(x', \pi)) - f^\pi(x, a)$$

where the last step is because of the definition of $\bar{f}_g^\pi$. For $x \in \mathcal{X}^+$, we have $x \in \mathcal{X}^+$ and the reward is zero, so

$$\bar{R}(x,a) + \gamma \bar{f}_g^\pi(x',\bar{\pi}) - \bar{f}_g^\pi(x,a) = 0$$

Therefore, we can derive

$(1-\gamma)Q^\pi(x_0,\pi) - (1-\gamma)\bar{f}_g^\pi(x_0,\bar{\pi})$
$\leq \mathbb{E}_{\bar{d}^{\bar{\pi}}}[\gamma \max(g^\pi(x'), f^\pi(x',\pi)) - f^\pi(x,a)|x \notin G, x \notin \mathcal{X}^+] + \mathbb{E}_{\bar{d}^{\bar{\pi}}}[R(x) - g^\pi(x)|x \in G]$

Finally, using Lemma A.2 we can have the final upper bound. $\qquad\square$

**Main Result: Performance Bound.** Let $\pi^\dagger$ be the learned policy and let $\bar{f}_g^{\pi^\dagger}$ be the learned function approximators. For any comparator policy $\pi$, let $\bar{f}_g^\pi = (f^\pi, g^\pi)$ be the estimator of $\pi$ on the data. We have.

$V^\pi(d_0) - V^{\pi^\dagger}(d_0)$
$= Q^\pi(d_0,\pi) - Q^{\pi^\dagger}(d_0,\pi^\dagger)$
$= Q^\pi(d_0,\pi) - \bar{f}_g^{\pi^\dagger}(d_0,\bar{\pi}^\dagger) + \bar{f}_g^{\pi^\dagger}(d_0,\bar{\pi}^\dagger) - Q^{\pi^\dagger}(d_0,\pi^\dagger)$
$\leq Q^\pi(d_0,\pi) - \bar{f}_g^{\pi^\dagger}(d_0,\bar{\pi}^\dagger)$
$\leq Q^\pi(d_0,\pi) - \bar{f}_g^\pi(d_0,\bar{\pi})$
$\leq \mathbb{E}_{\pi,P}\left[\sum_{t=0}^{T-1} \gamma^t(\gamma \max(g^\pi(x_{t+1}), f^\pi(x_{t+1},\pi)) - f^\pi(x_t,a_t)) + \gamma^T(R(x_T) - g^\pi(x_T))\right]$
$\leq \mathbb{E}_{\pi,P}\left[\sum_{t=0}^{T-1} \gamma^t|\gamma \max(g^\pi(x_{t+1}), f^\pi(x_{t+1},\pi)) - f^\pi(x_t,a_t)| + \gamma^T|R(x_T) - g^\pi(x_T)|\right]$
$\leq \mathfrak{C}_{\text{dyn}}(\pi)\mathbb{E}_{\mu_{\text{dyn}}}[|\gamma \max(g^\pi(x'), f^\pi(x',\pi)) - f^\pi(x,a)|] + \mathfrak{C}_{\text{goal}}(\pi)\mathbb{E}_{\mu_{\text{goal}}}[|g(x)-1|]$
$\lesssim \mathfrak{C}_{\text{dyn}}(\pi)\sqrt{\epsilon_{\text{dyn}}} + +\mathfrak{C}_{\text{goal}}(\pi)\sqrt{\epsilon_{\text{goal}}}$

where $\mathfrak{C}_{\text{dyn}}(\pi)$ and $\mathfrak{C}_{\text{goal}}(\pi)$ are the concentrability coefficients defined in Definition 5.3.

**Theorem A.11.** *Let $\pi^\dagger$ denote the learned policy of CODA + PSPI with datasets $D_{dyn}$ and $D_{goal}$, using value function classes $\mathcal{F} = \{\mathcal{X} \times \mathcal{A} \to [0,1]\}$ and $\mathcal{G} = \{\mathcal{X} \to [0,1]\}$. Under realizability and completeness assumptions as stated in Assumption 5.1 and Assumption 5.2 respectively, with probability $1 - \delta$, it holds, for any $\pi \in \Pi$,*

$$J(\pi) - J(\pi^\dagger) \lesssim \mathfrak{C}_{dyn}(\pi)\left(\sqrt{\frac{\square}{|D_{dyn}|}} + \sqrt{\frac{\square}{|D_{goal}|}}\right) + \mathfrak{C}_{goal}(\pi)\sqrt{\frac{\log \frac{\mathcal{N}_\infty(\mathcal{G},1/|D_{goal}|)}{\delta}}{|D_{goal}|}}$$

*where $\square \equiv \log\left(\frac{\mathcal{N}_\infty(\mathcal{F},1/|D_{goal}||D_{dyn}|)\mathcal{N}_\infty(\mathcal{G},1/|D_{goal}||D_{dyn}|)\mathcal{N}_{\infty,1}(\Pi,1/|D_{goal}||D_{dyn}|)}{\delta}\right)$, and $\mathfrak{C}_{dyn}(\pi)$ and $\mathfrak{C}_{goal}(\pi)$ are concentrability coefficients which decrease as the data coverage increases.*

### A.4.1 Proof of Lemmas A.12 and A.13

We first show the following complementary lemma where we use a concentration bound on the constructed datasets $\bar{D}_{\text{dyn}}$ and $\bar{D}_{\text{goal}}$. Lemmas A.7 and A.8 will follow deterministically from this main auxiliary result.

**Lemma A.12.** *With probability at least $1 - \delta$, for any $f, f_1, f_2 \in \mathcal{F}$ and $g \in \mathcal{G}$, we have:*

$\ell_{\bar{\mu}_{dyn}}(f_1, \bar{f}_g, \bar{\pi}) - \ell_{\bar{\mu}_{dyn}}(f_2, \bar{f}_g, \bar{\pi}) - \ell_{dyn}(f_1, \bar{f}_g, \bar{\pi}) + \ell_{dyn}(f_2, \bar{f}_g, \bar{\pi})$

$\leq \mathcal{O}\left(\|f_1 - f_2\|_{\bar{\mu}_{dyn}}\left(\sqrt{\frac{\square}{|D_{goal}|}} + \sqrt{\frac{\square}{|D_{dyn}|}}\right) + \frac{\square}{\sqrt{|D_{goal}||D_{dyn}|}} + \frac{\square}{|D_{goal}|} + \frac{\square}{|D_{dyn}|}\right)$

*where $\square \equiv \log\left(\frac{\mathcal{N}_\infty(\mathcal{F},1/|D_{goal}||D_{dyn}|)\mathcal{N}_\infty(\mathcal{G},1/|D_{goal}||D_{dyn}|)\mathcal{N}_{\infty,1}(\Pi,1/|D_{goal}||D_{dyn}|)}{\delta}\right)$.*

*Proof.* Our proof is similar to proof of corresponding results in Xie et al. [37] (Lemma A.4) and Cheng et al. [4] (Lemma 10) but we derive the result for the product distribution $\bar{\mu}_{\text{dyn}} = \mu_{\text{dyn}} \times \mu_{\text{goal}}$ and its empirical approximation using $\bar{D}_{\text{dyn}}$. Throughout this proof, we omit the bar on $\bar{\pi}$ as $\ell_{\text{dyn}}$ does not use the extended definition of the policy $\pi$ and further use $M, N$ for the dataset sizes $|D_{\text{goal}}|, |D_{\text{dyn}}|$. For any observed context $(c_j, s_j) \in D_{\text{goal}}$, we define the following quantity:

$$\ell^j_{\mu_{\text{dyn}}}(f, \bar{f}'_{g'}, \pi) = \mathbb{E}_{(s,a,s') \sim \mu_{\text{dyn}}} \left[ (f((s, c_j), a) - \gamma \max(g'((s', c_j))), f'((s', c_j), \pi)))^2 \right]$$

For conciseness, we use notation $x_{\circ j}$ for $(s, c_j)$ and $x'_{\circ j}$ for $(s', c_j)$ where $(s, a, s')$ is sampled from a dynamics distribution and $c_j \in D_{\text{goal}}$. We first start with the following:

$$\ell_{\bar{\mu}_{\text{dyn}}}(f_1, \bar{f}_g, \pi) - \ell_{\bar{\mu}_{\text{dyn}}}(f_2, \bar{f}_g, \pi) - \ell_{\text{dyn}}(f_1, \bar{f}_g, \pi) + \ell_{\text{dyn}}(f_2, \bar{f}_g, \pi)$$

$$\leq \ell_{\bar{\mu}_{\text{dyn}}}(f_1, \bar{f}_g, \pi) - \ell_{\bar{\mu}_{\text{dyn}}}(f_2, \bar{f}_g, \pi) - \frac{1}{M} \sum_{j=1}^{M} \ell^j_{\mu_{\text{dyn}}}(f_1, \bar{f}_g, \pi) + \frac{1}{M} \sum_{j=1}^{M} \ell^j_{\mu_{\text{dyn}}}(f_2, \bar{f}_g, \pi) \quad (11)$$

$$+ \sum_{j=1}^{M} \ell^j_{\mu_{\text{dyn}}}(f_1, \bar{f}_g, \pi) - \sum_{j=1}^{M} \ell^j_{\mu_{\text{dyn}}}(f_2, \bar{f}_g, \pi) - \ell_{\text{dyn}}(f_1, \bar{f}_g, \pi) + \ell_{\text{dyn}}(f_2, \bar{f}_g, \pi) \quad (12)$$

We will derive the final deviation bound by bounding each of these two empirical deviations in lines equation 11, equation 12. First, we will bound the term in line equation 11:

$$\sum_{j=1}^{M} \ell^j_{\mu_{\text{dyn}}}(f_1, \bar{f}_g, \pi) - \sum_{j=1}^{M} \ell^j_{\mu_{\text{dyn}}}(f_2, \bar{f}_g, \pi)$$

$$= \sum_{j=1}^{M} \ell^j_{\mu_{\text{dyn}}}(f_1, \bar{f}_g, \pi) - \ell^j_{\mu_{\text{dyn}}}(f_2, \bar{f}_g, \pi)$$

$$= \sum_{j=1}^{M} \mathbb{E}_{\mu_{\text{dyn}}} \left[ (f_1(x_{\circ j}, a) - \gamma \max(g(x'_{\circ j}), f(x'_{\circ j}, \pi)))^2 - (f_2(x_{\circ j}, a) - \gamma \max(g(x'_{\circ j}), f(x'_{\circ j}, \pi)))^2 \right]$$

$$= \sum_{j=1}^{M} \mathbb{E}_{\mu_{\text{dyn}}} \left[ (f_1(x_{\circ j}, a) - f_2(x_{\circ j}, a))(f_1(x_{\circ j}, a) + f_2(x_{\circ j}, a) - 2\gamma \max(g(x'_{\circ j}), f(x'_{\circ j}, \pi))) \right]$$

$$= \sum_{j=1}^{M} \mathbb{E}_{(s,a,\cdot) \sim \mu_{\text{dyn}}} \left[ (f_1(x_{\circ j}, a) - f_2(x_{\circ j}, a))(f_1(x_{\circ j}, a) + f_2(x_{\circ j}, a) - 2\bar{\mathcal{T}}^\pi \bar{f}_g)(x_{\circ j}, a)) \right]$$

$$(13)$$

$$= \sum_{j=1}^{M} \mathbb{E}_{(s,a,\cdot) \sim \mu_{\text{dyn}}} \left[ (f_1(x_{\circ j}, a) - \bar{\mathcal{T}}^\pi \bar{f}_g(x_{\circ j}, a))^2 - (f_2(x_{\circ j}, a) - \bar{\mathcal{T}}^\pi \bar{f}_g(x_{\circ j}, a))^2 \right]$$

Using a similar argument, we can show that:

$$\ell_{\bar{\mu}_{\text{dyn}}}(f_1, \bar{f}_g, \pi) - \ell_{\bar{\mu}_{\text{dyn}}}(f_2, \bar{f}_g, \pi)$$
$$= \mathbb{E}_{\bar{\mu}_{\text{dyn}}} \left[ (f_1((s, c), a) - \bar{\mathcal{T}}^\pi \bar{f}_g((s, c), a))^2 - (f_2((s, c), a) - \bar{\mathcal{T}}^\pi \bar{f}_g((s, c), a))^2 \right] \quad (14)$$

Let $\mathcal{F}_\epsilon$, $\mathcal{G}_\epsilon$ be $\epsilon$-cover of $\mathcal{F}$ and $\mathcal{G}$, and $\Pi_\epsilon$ be $\epsilon$-cover of $\Pi$, i.e., $\exists \tilde{f}_1, \tilde{f}_2, \tilde{f} \in \mathcal{F}_\epsilon$, $\tilde{g} \in GG_\epsilon$ and $\tilde{\pi} \in \Pi_\epsilon$ such that $\|f - \tilde{f}\|_\infty, \|f_1 - \tilde{f}_1\|_\infty, \|f_2 - \tilde{f}_2\|_\infty \leq \epsilon$ and $\|\pi\tilde{\pi}\|_{\infty,1} \leq \epsilon$.

Then, for any $f, f_1, f_2 \in \mathcal{F}, g \in \mathcal{G}, \pi \in \Pi$ and their corresponding $\tilde{f}, \tilde{f}_1, \tilde{f}_2 \in \mathcal{F}_\epsilon, \tilde{g} \in \mathcal{G}_\epsilon, \tilde{\pi} \in \Pi_\epsilon$:

$$\ell_{\bar{\mu}_{\mathrm{dyn}}}(\tilde{f}_1, \bar{\tilde{f}}_{\tilde{g}}, \tilde{\pi}) - \ell_{\bar{\mu}_{\mathrm{dyn}}}(\tilde{f}_2, \bar{\tilde{f}}_{\tilde{g}}, \tilde{\pi}) - \frac{1}{M}\sum_{j=1}^{M}\left(\ell^j_{\mu_{\mathrm{dyn}}}(\tilde{f}_1, \bar{\tilde{f}}_{\tilde{g}}, \tilde{\pi}) - \ell^j_{\mu_{\mathrm{dyn}}}(\tilde{f}_2, \bar{\tilde{f}}_{\tilde{g}}, \tilde{\pi})\right)$$

$$= \mathbb{E}_{\bar{\mu}_{\mathrm{dyn}}}\left[(\tilde{f}_1((s,c),a) - \bar{\mathcal{T}}^{\tilde{\pi}}\bar{\tilde{f}}_{\tilde{g}}((s,c),a))^2 - (\tilde{f}_2((s,c),a) - \bar{\mathcal{T}}^{\tilde{\pi}}\bar{\tilde{f}}_{\tilde{g}}((s,c),a))^2\right]$$

$$- \frac{1}{M}\sum_{j=1}^{M}\mathbb{E}_{(s,a,\cdot)\sim\mu_{\mathrm{dyn}}}\left[(\tilde{f}_1(x_{\circ j},a) - \tilde{f}_2(x_{\circ j},a))(\tilde{f}_1(x_{\circ j},a) + \tilde{f}_2(x_{\circ j},a) - 2\bar{\mathcal{T}}^{\tilde{\pi}}\bar{\tilde{f}}_{\tilde{g}})\right]$$

$$\leq \sqrt{\frac{4\mathbf{V}\log\left(\frac{\mathcal{N}_\infty(\mathcal{F},\epsilon)\mathcal{N}_\infty(\mathcal{G},\epsilon)\mathcal{N}_{\infty,1}(\Pi,\epsilon)}{\delta}\right)}{M}} + \frac{2\log\left(\frac{\mathcal{N}_\infty(\mathcal{F},\epsilon)\mathcal{N}_\infty(\mathcal{G},\epsilon)\mathcal{N}_{\infty,1}(\Pi,\epsilon)}{\delta}\right)}{3M}.$$

where the first equation follows from eqs. equation 13 and equation 14, and the last inequality follows from Bernstein's inequality with a union bound over the classes $\mathcal{F}_\epsilon, \mathcal{G}_\epsilon, \Pi_\epsilon$ where $\mathbf{V}$ is the variance term as follows:

$$\mathrm{Var}_{c\sim\mu_{\mathrm{goal}}}\left[\mathbb{E}_{(s,a,\cdot)\sim\mu_{\mathrm{dyn}}}\left[(f_1((s,c),a) - f_2((s,c),a))(f_1((s,c),a) + f_2((s,c),a) - 2\bar{\mathcal{T}}^\pi\bar{f}_g((s,c),a))\right]\right]$$

$$\leq \mathbb{E}_{c\sim\mu_{\mathrm{goal}}}\left[\mathbb{E}_{(s,a,\cdot)\sim\mu_{\mathrm{dyn}}}\left[(f_1((s,c),a) - f_2((s,c),a))(f_1((s,c),a) + f_2((s,c),a) - 2\bar{\mathcal{T}}^\pi\bar{f}_g((s,c),a))\right]^2\right]$$

$$\leq 4\mathbb{E}_{\bar{\mu}_{\mathrm{dyn}}}\left[(f_1((s,c),a) - f_2((s,c),a))^2\right]$$

where we used that fact that $f, g \in [0,1]$.

Thus, with probability $1 - \delta$,

$$\ell_{\bar{\mu}_{\mathrm{dyn}}}(\tilde{f}_1, \bar{\tilde{f}}_{\tilde{g}}, \tilde{\pi}) - \ell_{\bar{\mu}_{\mathrm{dyn}}}(\tilde{f}_2, \bar{\tilde{f}}_{\tilde{g}}, \tilde{\pi}) - \frac{1}{M}\sum_{j=1}^{M}\left(\ell^j_{\mu_{\mathrm{dyn}}}(\tilde{f}_1, \bar{\tilde{f}}_{\tilde{g}}, \tilde{\pi}) - \ell^j_{\mu_{\mathrm{dyn}}}(\tilde{f}_2, \bar{\tilde{f}}_{\tilde{g}}, \tilde{\pi})\right)$$

$$\leq 2\|\tilde{f}_1 - \tilde{f}_2\|_{\bar{\mu}_{\mathrm{dyn}}}\sqrt{\frac{\log\left(\frac{\mathcal{N}_\infty(\mathcal{F},\epsilon)\mathcal{N}_\infty(\mathcal{G},\epsilon)\mathcal{N}_{\infty,1}(\Pi,\epsilon)}{\delta}\right)}{M}} + \frac{2\log\left(\frac{\mathcal{N}_\infty(\mathcal{F},\epsilon)\mathcal{N}_\infty(\mathcal{G},\epsilon)\mathcal{N}_{\infty,1}(\Pi,\epsilon)}{\delta}\right)}{3M}.$$

Using the property of the set covers of $\mathcal{F}, \mathcal{G}, \Pi$, we can easily conclude that:

$$\ell_{\bar{\mu}_{\mathrm{dyn}}}(f_1, \bar{f}_g, \pi) - \ell_{\bar{\mu}_{\mathrm{dyn}}}(f_2, \bar{f}_g, \pi) - \frac{1}{M}\sum_{j=1}^{M}\left(\ell^j_{\mu_{\mathrm{dyn}}}(f_1, \bar{f}_g, \pi) - \ell^j_{\mu_{\mathrm{dyn}}}(f_2, \bar{f}_g, \pi)\right)$$

$$\lesssim \|f_1 - f_2\|_{\bar{\mu}_{\mathrm{dyn}}}\sqrt{\frac{\log\left(\frac{\mathcal{N}_\infty(\mathcal{F},\epsilon)\mathcal{N}_\infty(\mathcal{G},\epsilon)\mathcal{N}_{\infty,1}(\Pi,\epsilon)}{\delta}\right)}{M}} + \frac{\log\left(\frac{\mathcal{N}_\infty(\mathcal{F},\epsilon)\mathcal{N}_\infty(\mathcal{G},\epsilon)\mathcal{N}_{\infty,1}(\Pi,\epsilon)}{\delta}\right)}{M}$$

$$+ \epsilon\sqrt{\frac{\log\left(\frac{\mathcal{N}_\infty(\mathcal{F},\epsilon)\mathcal{N}_\infty(\mathcal{G},\epsilon)\mathcal{N}_{\infty,1}(\Pi,\epsilon)}{\delta}\right)}{M}} + \epsilon. \tag{15}$$

Now, we bound the second deviation term in eq. line equation 12:

$$\frac{1}{M}\sum_{j=1}^{M}\left(\ell^j_{\mu_{\mathrm{dyn}}}(f_1, \bar{f}_g, \pi) - \ell^j_{\mu_{\mathrm{dyn}}}(f_2, \bar{f}_g, \pi)\right) - \left(\ell_{\mathrm{dyn}}(f_1, \bar{f}_g, \pi) - \ell_{\mathrm{dyn}}(f_2, \bar{f}_g, \pi)\right)$$

$$= \frac{1}{M}\sum_{j=1}^{M}\Bigg[\mathbb{E}_{\mu_{\mathrm{dyn}}}\left[(f_1(x_{\circ j},a) - \gamma\max(g(x'_{\circ j}), f(x'_{\circ j}, \pi)))^2 - (f_2(x_{\circ j},a) - \gamma\max(g(x'_{\circ j}), f(x'_{\circ j}, \pi)))^2\right]$$

$$- \frac{1}{N}\sum_{i=1}^{N}\left[(f_1(x_{ij},a) - \gamma\max(g(x'_{ij}), f(x'_{ij}, \pi)))^2 - (f_2(x_{ij},a) - \gamma\max(g(x'_{ij}), f(x'_{ij}, \pi)))^2\right]\Bigg]$$

$$\tag{16}$$

For any fixed $c_j$, using the same strategy as we used for bounding the first term in eq. line equation 11, for any $f, f_1, f_2 \in \mathcal{F}$, $g \in \mathcal{G}$, $\pi \in \Pi$ and their corresponding $\tilde{f}, \tilde{f}_1, \tilde{f}_2 \in \mathcal{F}_\epsilon$, $\tilde{g} \in \mathcal{G}_\epsilon$, $\tilde{\pi} \in \Pi_\epsilon$, with probability at least $1 - \delta$:

$$\left( \ell^j_{\mu_{\mathrm{dyn}}}(\tilde{f}_1, \bar{\tilde{f}}_{\tilde{g}}, \tilde{\pi}) - \ell^j_{\mu_{\mathrm{dyn}}}(\tilde{f}_2, \bar{\tilde{f}}_{\tilde{g}}, \tilde{\pi}) \right) - \left( \ell_{\mathrm{dyn}}(\tilde{f}_1, \bar{\tilde{f}}_{\tilde{g}}, \tilde{\pi}) - \ell_{\mathrm{dyn}}(\tilde{f}_2, \bar{\tilde{f}}_{\tilde{g}}, \tilde{\pi}) \right)$$

$$\lesssim \|\tilde{f}_1 - \tilde{f}_2\|_{\mu_{\mathrm{dyn}} \times \{c_j\}} \sqrt{\frac{\log\left( \frac{\mathcal{N}_\infty(\mathcal{F},\epsilon)\mathcal{N}_\infty(\mathcal{G},\epsilon)\mathcal{N}_{\infty,1}(\Pi,\epsilon)}{\delta} \right)}{N}} + \frac{\log\left( \frac{\mathcal{N}_\infty(\mathcal{F},\epsilon)\mathcal{N}_\infty(\mathcal{G},\epsilon)\mathcal{N}_{\infty,1}(\Pi,\epsilon)}{\delta} \right)}{N}$$

$$+ \epsilon \sqrt{\frac{\log\left( \frac{\mathcal{N}_\infty(\mathcal{F},\epsilon)\mathcal{N}_\infty(\mathcal{G},\epsilon)\mathcal{N}_{\infty,1}(\Pi,\epsilon)}{\delta} \right)}{N}} + \epsilon.$$

We can now consider the sum in the second term in eq. line equation 12 for $\tilde{f}, \tilde{f}_1, \tilde{f}_2, \tilde{g}, \tilde{\pi}$ as:

$$\frac{1}{M} \sum_{j=1}^{M} \left( \ell^j_{\mu_{\mathrm{dyn}}}(\tilde{f}_1, \bar{\tilde{f}}_{\tilde{g}}, \tilde{\pi}) - \ell^j_{\mu_{\mathrm{dyn}}}(\tilde{f}_2, \bar{\tilde{f}}_{\tilde{g}}, \tilde{\pi}) \right) - \left( \ell_{\mathrm{dyn}}(\tilde{f}_1, \bar{\tilde{f}}_{\tilde{g}}, \tilde{\pi}) - \ell_{\mathrm{dyn}}(\tilde{f}_2, \bar{\tilde{f}}_{\tilde{g}}, \tilde{\pi}) \right)$$

$$\lesssim \frac{1}{M} \sum_{j=1}^{M} \|\tilde{f}_1 - \tilde{f}_2\|_{\mu_{\mathrm{dyn}} \times \{c_j\}} \sqrt{\frac{\log\left( \frac{\mathcal{N}_\infty(\mathcal{F},\epsilon)\mathcal{N}_\infty(\mathcal{G},\epsilon)\mathcal{N}_{\infty,1}(\Pi,\epsilon)}{\delta} \right)}{N}}$$

$$+ \frac{\log\left( \frac{\mathcal{N}_\infty(\mathcal{F},\epsilon)\mathcal{N}_\infty(\mathcal{G},\epsilon)\mathcal{N}_{\infty,1}(\Pi,\epsilon)}{\delta} \right)}{N} + \epsilon \sqrt{\frac{\log\left( \frac{\mathcal{N}_\infty(\mathcal{F},\epsilon)\mathcal{N}_\infty(\mathcal{G},\epsilon)\mathcal{N}_{\infty,1}(\Pi,\epsilon)}{\delta} \right)}{N}} + \epsilon.$$

$$\lesssim \left( \frac{1}{M} \sum_{j=1}^{M} \|\tilde{f}_1 - \tilde{f}_2\|_{\mu_{\mathrm{dyn}} \times \{c_j\}} - \|\tilde{f}_1 - \tilde{f}_2\|_{\bar{\mu}_{\mathrm{dyn}}} \right) \sqrt{\frac{\log\left( \frac{\mathcal{N}_\infty(\mathcal{F},\epsilon)\mathcal{N}_\infty(\mathcal{G},\epsilon)\mathcal{N}_{\infty,1}(\Pi,\epsilon)}{\delta} \right)}{N}}$$

$$+ \|\tilde{f}_1 - \tilde{f}_2\|_{\bar{\mu}_{\mathrm{dyn}}} \sqrt{\frac{\log\left( \frac{\mathcal{N}_\infty(\mathcal{F},\epsilon)\mathcal{N}_\infty(\mathcal{G},\epsilon)\mathcal{N}_{\infty,1}(\Pi,\epsilon)}{\delta} \right)}{N}}$$

$$+ \frac{\log\left( \frac{\mathcal{N}_\infty(\mathcal{F},\epsilon)\mathcal{N}_\infty(\mathcal{G},\epsilon)\mathcal{N}_{\infty,1}(\Pi,\epsilon)}{\delta} \right)}{N} + \epsilon \sqrt{\frac{\log\left( \frac{\mathcal{N}_\infty(\mathcal{F},\epsilon)\mathcal{N}_\infty(\mathcal{G},\epsilon)\mathcal{N}_{\infty,1}(\Pi,\epsilon)}{\delta} \right)}{N}} + \epsilon.$$

$$\lesssim \|\tilde{f}_1 - \tilde{f}_2\|_{\bar{\mu}_{\mathrm{dyn}}} \sqrt{\frac{\log\left( \frac{\mathcal{N}_\infty(\mathcal{F},\epsilon)\mathcal{N}_\infty(\mathcal{G},\epsilon)\mathcal{N}_{\infty,1}(\Pi,\epsilon)}{\delta} \right)}{N}} + \frac{\log\left( \frac{\mathcal{N}_\infty(\mathcal{F},\epsilon)\mathcal{N}_\infty(\mathcal{G},\epsilon)\mathcal{N}_{\infty,1}(\Pi,\epsilon)}{\delta} \right)}{\sqrt{NM}}$$

$$+ \frac{\log\left( \frac{\mathcal{N}_\infty(\mathcal{F},\epsilon)\mathcal{N}_\infty(\mathcal{G},\epsilon)\mathcal{N}_{\infty,1}(\Pi,\epsilon)}{\delta} \right)}{N} + \epsilon \sqrt{\frac{\log\left( \frac{\mathcal{N}_\infty(\mathcal{F},\epsilon)\mathcal{N}_\infty(\mathcal{G},\epsilon)\mathcal{N}_{\infty,1}(\Pi,\epsilon)}{\delta} \right)}{N}} + \epsilon.$$

$$(17)$$

where the last inequality follows from Hoeffding's inequality. We can now bound the term in eq. line equation 12 as:

$$\frac{1}{M} \sum_{j=1}^{M} \left( \ell^j_{\mu_{\mathrm{dyn}}}(f_1, \bar{f}_g, \pi) - \ell^j_{\mu_{\mathrm{dyn}}}(f_2, \bar{f}_g, \pi) \right) - \left( \ell_{\mathrm{dyn}}(f_1, \bar{f}_g, \pi) - \ell_{\mathrm{dyn}}(f_2, \bar{f}_g, \pi) \right)$$

$$\lesssim \|\tilde{f}_1 - \tilde{f}_2\|_{\bar{\mu}_{\mathrm{dyn}}} \sqrt{\frac{\log\left( \frac{\mathcal{N}_\infty(\mathcal{F},\epsilon)\mathcal{N}_\infty(\mathcal{G},\epsilon)\mathcal{N}_{\infty,1}(\Pi,\epsilon)}{\delta} \right)}{N}} + \frac{\log\left( \frac{\mathcal{N}_\infty(\mathcal{F},\epsilon)\mathcal{N}_\infty(\mathcal{G},\epsilon)\mathcal{N}_{\infty,1}(\Pi,\epsilon)}{\delta} \right)}{\sqrt{NM}}$$

$$+ \frac{\log\left( \frac{\mathcal{N}_\infty(\mathcal{F},\epsilon)\mathcal{N}_\infty(\mathcal{G},\epsilon)\mathcal{N}_{\infty,1}(\Pi,\epsilon)}{\delta} \right)}{N} + \epsilon \sqrt{\frac{\log\left( \frac{\mathcal{N}_\infty(\mathcal{F},\epsilon)\mathcal{N}_\infty(\mathcal{G},\epsilon)\mathcal{N}_{\infty,1}(\Pi,\epsilon)}{\delta} \right)}{N}} + \epsilon. \qquad (18)$$

Combining eqs. equation 15 and equation 18 with $\epsilon = \mathcal{O}(\frac{1}{MN})$, we get the final result. $\square$

**Lemma A.13.** *With probability at least $1 - \delta$, for any $g, g_1, g_2 \in \mathcal{G}$ and $f \in \mathcal{F}$, we have:*

$$\ell_{\bar{\mu}_{goal}}(\bar{f}_{g_1}) - \ell_{\bar{\mu}_{goal}}(\bar{f}_{g_2}) - \ell_{goal}(\bar{f}_{g_1}) + \ell_{goal}(\bar{f}_{g_2})$$

$$\leq \mathcal{O}\left( \|g_1 - g_2\|_{\bar{\mu}_{goal}} \sqrt{\frac{\log\left(\frac{\mathcal{N}_\infty(\mathcal{F}, 1/|D_{goal}|)}{\delta}\right)}{|D_{goal}|}} + \frac{\log\left(\frac{\mathcal{N}_\infty(\mathcal{F}, 1/|D_{goal}|)}{\delta}\right)}{|D_{goal}|} \right).$$

*Proof.* This result can be proven using the same arguments as used in Lemma A.12 using a covering argument just over $\mathcal{G}$. $\square$

Using these two main concentration results, we can now prove Lemmas A.7 and A.8.

***Proof of Lemma A.7.*** Note $\bar{Q}^{\bar{\pi}} = \bar{f}_g$ for some $f \in \mathcal{F}$ and $g \in \mathcal{G}$ (Proposition A.6) and

$$0 = \mathbb{E}_{x, a \sim \mu_{dyn}}[(\bar{Q}^{\bar{\pi}}(x, a) - \bar{\mathcal{T}}^{\bar{\pi}} \bar{Q}^{\bar{\pi}}(x, a))^2]$$

$$= \mathbb{E}_{x, a \sim \mu_{dyn}}[(\bar{Q}^{\bar{\pi}}(x, a) - 0 - \gamma \mathbb{E}_{x' \sim \bar{P}(\cdot|x, a)}[\phi(\bar{Q}^{\bar{\pi}})(x', \pi)])^2]$$

The lemma can now be proved by following a similar proof of Theorem A.1 of Xie et al. [37]. The key difference is the use of our concentration bounds in Lemmas A.12 and A.13 instead of Lemma A.4 in the proof of Xie et al. [37]. On the other hand, $\ell_{goal}(\bar{f}_g) = 0$ because the reward $R(x)$ is deterministic which results in the second inequality. $\square$

***Proof of Lemma A.8.*** This result can again be proved using the same steps as in Lemma A.5 from Xie et al. [37] based on the concentration bound in Lemmas A.12 and A.13. $\square$

# B  Experimental Details

## B.1  Hyperparameters and Experimental Settings

**IQL.** For IQL, we keep the hyperparameter of $\gamma = 0.99$, $\tau = 0.9$, $\beta = 10.0$, and $\alpha = 0.005$ in [18], and tune other hyperparameters on the antmaze-medium-play-v2 environment and choose batch size = 1024 from candidate choices $\{256, 512, 1024, 2046\}$, learning rate = $10^{-4}$ from candidate choices $\{5 \cdot 10^{-5}, 10^{-4}, 3 \cdot 10^{-4}\}$ and 3 layer MLP with RuLU activating and 256 hidden units for all networks. We use the same set of IQL hyperparameters for both our methods and all the baseline methods included in Section 6.2, and apply it to all environments. In the experiments, we follow the convention of the $-1/0$ reward in the IQL implementation for Antmaze, which can be shown to be the same as the $0/1$ reward notion in terms of ranking policies under the discounted MDP setting.

**Reward Prediction (RP).** For naive reward prediction, we use the full context-goal dataset as positive data, and train a reward model with 3-layer MLP and ReLU activations, learning rate = $10^{-4}$, batch size = 1024, and training for 100 epochs for convergence. To label the transition dataset, we need to find some appropriate threshold to label states predicted as goals given contexts. We choose the percentile as 5% in the reward distribution evaluated by the context-goal set as the threshold to label goals (if a reward is larger than the threshold than it is labeled as terminal), from candidate choices $\{0\%, 5\%, 10\%\}$. Then we apply it to all environments. Another trick we apply for the reward prediction is that instead of predicting 0 for the context-goal dataset, we let it predict 1 but shift the reward prediction by -1 during reward evaluation, which prevents the model from learning all 0 weights. Similar tricks are also used in other reward learning baselines.

**UDS+RP.** We use the same structure and training procedure for the reward model as RP, except that we also randomly sample a minibatch of "negative" contextual transitions with the same batch size for a balanced distribution, which is constructed by randomly sampling combinations of a state in the trajectory-only dataset and a context from the context-goal dataset. To create a balanced distribution of positive and negative samples, we sample from each dataset with equal probability. For the threshold, we choose the percentile as 5% in the reward distribution evaluated by the context-goal set as the threshold to label goals in the antmaze-medium-play-v2 environment, from candidate choices $\{0\%, 5\%, 10\%\}$. Then we apply it to all environments.

**PDS.** We use the same structure and training procedure for the reward model as RP, except that we train an ensemble of 10 networks as in [14]. To select the threshold percentile and the pessimistic weight $k$, we choose the percentile as 15% in the reward distribution evaluated by the context-goal set as the threshold to label goals from candidate choices $\{0\%, 5\%, 10\%, 15\%, 20\%\}$, and $k = 15$ from the candidate choices $\{5, 10, 15, 20\}$ in the antmaze-medium-play-v2 environment. Then we apply them to all environments.

**CODA (ours).** We do not require extra parameters other than the possibility of sampling from the real and fake transitions. Intuitively, we should sample from both datasets with the same probability to create an overall balanced distribution. We ran additional experiments to study the effect of this sampling ratio hyperparameter: ratio of samples from the context-goal dataset $D_{\text{goal}}$ to total samples in each minibatch. Table 5 shows that CODA well as long as the ratio is roughly balanced in sampling from both dataset.

**Compute Resources.** For all methods, each training run takes about 8h on a NVIDIA T4 GPU.

Table 5: Average success rate (%) in AntMaze-v2 from all environments, with different sampling ratios from the context-goal dataset.

| Env/Ratio | 0.1 | 0.3 | 0.5 | 0.7 | 0.9 |
|---|---|---|---|---|---|
| umaze | 91.6±1.3 | 92.4±1.0 | 94.8±1.3 | 86.4±1.8 | 84.8±3.0 |
| umaze diverse | 76.8±1.9 | 79.2±1.6 | 72.8±7.7 | 76.6±2.3 | 65.4±8.8 |
| medium play | 82.3±2.1 | 85.0±1.8 | 75.8±1.9 | 72.8±1.3 | 76.6±1.3 |
| medium diverse | 79.4±1.6 | 76.6±3.0 | 84.5±5.2 | 75.6±2.0 | 72.0±3.5 |
| large play | 50.8±2.0 | 45.2±3.7 | 60.0±7.6 | 43.6±2.3 | 46.6±2.3 |
| large diverse | 35.8±5.7 | 37.4±4.7 | 36.8±6.9 | 34.4±2.4 | 27.0±2.1 |
| average | 69.5 | 68.9 | 70.8 | 64.9 | 62.1 |

## B.2 Context-Goal dataset Construction and Environmental Evaluation.

Here we introduce the context-goal dataset in the three levels of context-goal setup mentioned in Section 6 and how to evaluate in each setup. We also include our code implementation for reference.

**Original Antmaze.** We extract the 2D locations from the states in the trajectory dataset with terminal=True as the context (in original antmaze, it suffices to reach the $L_2$ ball with radius 0.5 around the center), where the contexts are distributed very closely as visualized in Figure 2(a), and the corresponding states serve as the goal examples with Gaussian perturbations $N(0, 0.05)$ on the dimensions other than the 2D location.

**Four Rooms.** For each maze map, we partition 4 rooms like Figure 2(b) and use the room number as the context. To construct goal examples, we create a copy of all states in the trajectory dataset, perturb the states in the copy by $N(0, 0.05)$ on each dimension, and then randomly select the states (up to 20K) according to the room partition.

**Random Cells.** For each maze map, we construct a range of non-wall 2D locations in the maze map and uniformly sample from it to get the training contexts. To construct the goal set given context, we randomly sample up to 20K states with the 2D locations within the $L_2$ ball with radius 2. Figure 2(C) is a intuitive visualization of the corresponding context-goal sets. For test distributions, we have two settings: 1) the same as the training distribution; 2) test contexts are drawn from a limited area that is far away from the starting point of the agent.

**Evaluation.** We follow the conventional evaluation procedure in [18], where the success rate is normalized to be 0-100 and evaluated with 100 trajectories. We report the result with standard error across 5 random seeds. The oracle condition we define in each context-goal setup is used to evaluate whether the agent has successfully reached the goal and also defines the termination of an episode.

## B.3  Reward Model Evaluation

For reward learning baselines, we evaluate the learned reward model to showcase whether the learned reward function can successfully capture context-goal relationships.

**Evaluation dataset construction.**  We construct the positive dataset from context-goal examples, and the negative dataset from the combination of the context set and all states in the trajectory-only data, then use the oracle context-goal definition in each setup to filter out the positive ones. We then evaluate the predicted reward on both positive and negative datasets, generating boxplots to visualize the distributions of the predicted reward for both datasets.

**Results.**  Here we present boxplots for reward models with experimental setups in Section 6.3. Overall we observe that PDS+RP is consistently better at separating positive and negative distributions than UDS and naive reward prediction. However, PDS can still fail at fully separating positive and negative examples.

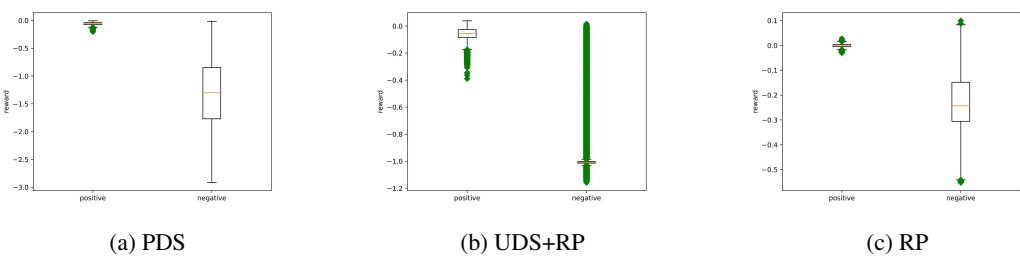

(a) PDS  (b) UDS+RP  (c) RP

Figure 3: Reward model evaluation for the large-diverse dataset for original AntMaze environment. Green dots are outliers.

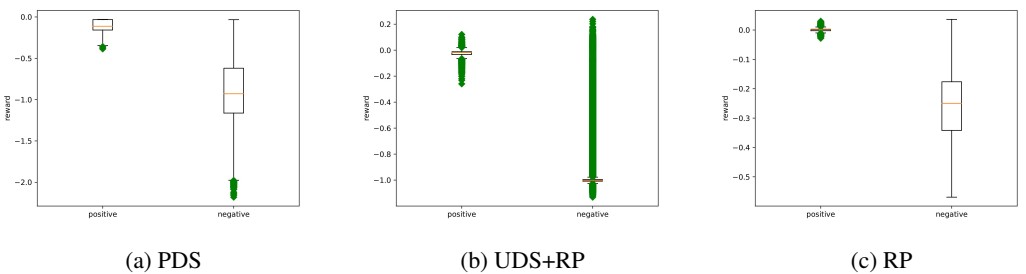

(a) PDS  (b) UDS+RP  (c) RP

Figure 4: Reward model evaluation for the medium-diverse dataset for the original AntMaze environment. Green dots are outliers.

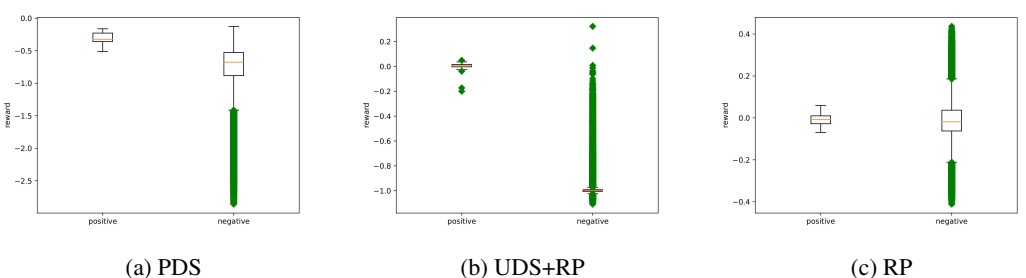

(a) PDS  (b) UDS+RP  (c) RP

Figure 5: Reward model evaluation for the umaze-diverse dataset for the original AntMaze environment. Green dots are outliers.

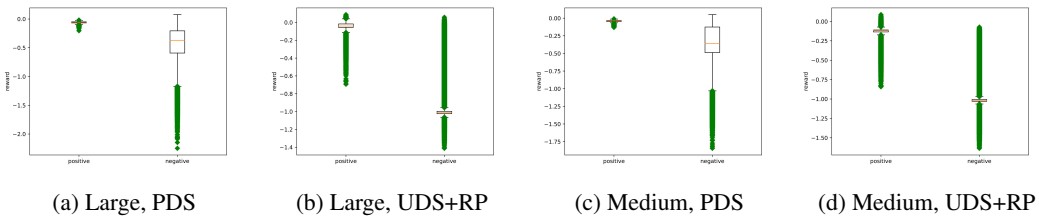

| (a) Large, PDS | (b) Large, UDS+RP | (c) Medium, PDS | (d) Medium, UDS+RP |

Figure 6: Reward model evaluation for the Four Rooms environment. Green dots are outliers.

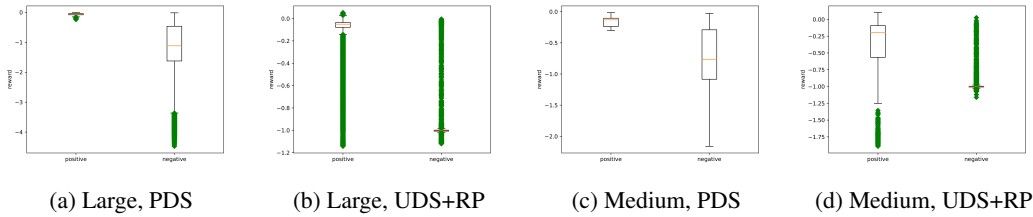

| (a) Large, PDS | (b) Large, UDS+RP | (c) Medium, PDS | (d) Medium, UDS+RP |

Figure 7: Reward evaluation for Random Cells environment (the test context distribution is the same as training). Green dots are outliers.

