# OpenReview forum: "How to Solve Contextual Goal-Oriented Problems with Offline Datasets?"
_NeurIPS.cc/2024/Conference — NeurIPS 2024 poster_

### Official Review · Reviewer_ug4M · 2024-07-04

**Soundness:** 4
**Presentation:** 4
**Contribution:** 4
**Rating:** 7
**Confidence:** 2

**Summary:**

This paper proposes a novel method to solve CGO problems and proves CODA can learn near-optimal policies without the need for negative labels with natural assumptions. In addition, sufficient experiments prove the effectiveness of the proposed method.

**Strengths:**

1. The contribution of the proposed method is interesting.
2. Sufficient experiments prove the effectiveness of the proposed method.
3. Theoretical analysis is sufficient.

**Weaknesses:**

None

**Questions:**

None

---

> ### Author Rebuttal · Authors · 2024-08-06
>
> Thanks for your review and the acknowledgement.

---

### Official Review · Reviewer_Kh9A · 2024-07-13

**Soundness:** 3
**Presentation:** 3
**Contribution:** 2
**Rating:** 5
**Confidence:** 3

**Summary:**

This paper focuses on a new RL task: Contextual Goal-Oriented Offline RL task. This task considers an offline goal-context pair dataset and an unsupervised transition dataset. To address this task, in this paper, Contextual goal-Oriented Data Augmentation (CODA) is proposed to augment a new state $s^+$ and new action $a^+$ representing the goal within the context is achieved with reward as 1. The theoretical analysis is proposed to demonstrate CODA's capability to solve CGO problems.

**Strengths:**

1) This work focuses on a new task where goal and context are related. I am not familiar with the literature of Contextual Goal-Oriented (CGO).  Personally, I think this task is interesting.

2) CODA methods show good performance on empirical evaluations.

**Weaknesses:**

1) In the Related Work section, some methods to solve Goal-conditioned RL are listed. Can these methods be used to solve this problem? If yes, how do these methods perform in the empirical studies?  What is the relation between these methods and the used baseline methods?

2)  This paper proposes a data augmentation method. Can this method be incorporated with other Goal-conditioned RL approaches to achieve better performance, such as relabeling?

3) The proposed data augmentation method is straightforward. It uses goal and context to label states and actions. Why such data augmentation can achieve such impressive results? More discussions are expected.

4) More datasets and real-world scenarios can be used to further demonstrate the effectiveness of this method.

**Questions:**

Some questions are listed in the Weaknesses.

The final score will be updated when these questions are answered.

**Limitations:**

There is no clear negative societal impact.

---

> ### Author Rebuttal · Authors · 2024-08-06
>
> We sincerely appreciate your valuable comments. We found them extremely helpful in improving our draft. We address each comment below.
>
> ### 1. “Can goal conditioned methods be used to solve this problem”:
>
> There is no straight-forward way to directly apply typical goal-conditioned methods to our offline CGO setup in general, since the relationship between contexts and goals is **unknown** here. On the other hand, the traditional GO setting (equivalently a CGO special case where context = goal) essentially assumes the relationship between contexts and goals is **known**; and algorithms like Hindsight Experience Replay (HER) critically relies on this assumption for the relabeling.
>
> Nonetheless, we still try to show how goal-conditioned RL methods might perform in this more general setting. We do this by incorporating a goal-prediction baseline, which first learns to predict goals given a context using the context-goal dataset, and then uses the goal-conditioned policy learned from the dynamics-only dataset with HER (L45). We presented the result of the “**Goal Prediction**” baseline in **Table 1, 2, 3**.
>
> ### 2. “Can this method be incorporated with other Goal-conditioned RL approaches to achieve better performance, such as relabeling”:
> As mentioned above, due to the missing relationship between goals and contexts, goal-conditioned RL methods cannot be directly applied. We do highlight HER is used in our Goal Prediction baseline to learn the goal-conditioned policy (**L307**).
>
> There is no way to directly relabel contexts in our offline CGO setup since the context is not available in the dynamics dataset, and we only have a fixed and limited context-goal dataset. Similarly, for other goal conditioned RL approaches, we cannot directly use that due to missing context labels for the states in the dynamics dataset.
>
> ### 3. Why such data augmentation can achieve such impressive results?
>
> As discussed in our introduction, the baseline methods all have their drawbacks (**L44**): For the goal prediction methods, the predicted goal might not always be feasible given the initial state. For the reward prediction baseline, it does not make good use of the goal-oriented nature of the CGO problems, and it might be challenging to infer reasonable reward for any context-goal pair, given only with positive data is available during training (context-goal dataset).
>
> On the other hand, in CODA, we carefully design the **augmented MDP** structure which our data augmentation method relies on. In this way, 1) **we do not need to deal with the challenges of predicting the feasible goals** in this augmented MDP like goal prediction methods and 2) **we do not need to handle the missing context label problem** in this augmented MDP. Notice that this augmented MDP **equivalent to the original MDP**. This is why this augmentation method is effective despite its simplicity: **it fully makes use of the CGO structure of the problem to circumvent the drawbacks in other baseline methods**.
>
> ### 4. More datasets and real-world scenarios:
>
> We acknowledge that having more datasets would be better. However, our main goal of this paper is to formulate the CGO setup since it hasn’t been formally studied in the literature, and to show it can be probably solved with commonly available offline datasets, as demonstrated by the proposed method. Our controlled experiments, although limited to the same simulator, are designed to cover the spectrum of different CGO setups listed in the taxonomy given in **Section 6 (L265)**. We think this is a sufficient first milestone to showcase the effectiveness of the proposed method. Testing on real world datasets is important future work but is out of the current scope of this paper.
>
> We hope our response resolves all the concerns in your review and please feel free to let us know if you have further questions. If our responses have addressed your concerns, please kindly consider raising the score. Thank you!

---

> > ### Author Response · Authors · 2024-08-10
> >
> > Dear reviewer, thank you again for the review and we hope our rebuttal address your concerns. We would greatly appreciate your feedback and please feel free to let us know if you have any other questions in the discussion period!

---

> > > ### Comment · Reviewer_Kh9A · 2024-08-12
> > > **Sorry for delay**
> > >
> > > I sincerely appreciate the authors' careful consideration of my concerns and their detailed responses. I apologize for the delayed reply.
> > >
> > > The connections between the proposed augmentation and other Goal-conditioned RL methods are now much clearer to me, and I will adjust my score accordingly. As I am not an expert in this specific field, my review is based on a general understanding of machine learning and academic writing. Given that all reviews, as noted by Reviewer CzDa, are of low confidence, I change to support acceptance but maintain a "borderline" rating.

---

### Official Review · Reviewer_ek4e · 2024-07-13

**Soundness:** 3
**Presentation:** 1
**Contribution:** 2
**Rating:** 4
**Confidence:** 2

**Summary:**

The paper combines an (unlabeled) dynamics dataset of trajectories, and a (labeled) context-goal dataset in an offline setting to create a combined MDP(Markov Decision Process). They do it by augmenting the dynamic dataset to have fallacious action to the terminal state with reward 1 on goal states given context c. All other states have a reward of 0. They also give a theoretical proof of their methodology.

**Strengths:**

The paper presents a general solution that can be applied to a wide range of problems within the field; the findings and methods have broad applicability, increasing their relevance and potential impact across various contexts.

The authors have validated their methodology through comprehensive proofs and empirical evidence (to some extent). This thorough validation provides a solid foundation.

**Weaknesses:**

-Limited novelty, the paper provides a way to combine two types of datasets, but the methodology it provides is trivial

-Paper does provide a mathematical proof of the technique, but the technique itself is straightforward enough that the proof is trivial given that it builds on baseline proof

-Not adequate experimentation

-Experimentation on a very simple problem set

-The paper needs more polishing (e.g., MDP acronym used in the abstract without initializing,  Line 8: "outperform" should be "outperforms"; Line 177: "fictious" should be "fictitious". etc.)

**Questions:**

See the weakness section

---

> ### Author Rebuttal · Authors · 2024-08-06
>
> Thank you for the review. We address each comment below.
>
> ### 1. Limited novelty:
> Could you provide specific reasons why the novelty is limited? We respectfully disagree that our method is trivial since we carefully design the augmented MDP structure and the data augmentation method such that 1) we do not need to deal with the challenges of predicting the feasible goals in this augmented MDP and 2) we do not need to handle the missing context label problem in this augmented MDP. This is why this augmentation method is effective despite its simplicity: **it fully makes use of the CGO structure of the problem to circumvent the drawbacks in other baseline methods. Simplicity is not a drawback and does not equal lack of novelty.**
>
> ### 2. Proof is trivial:
> A major aspect of our proposed algorithm is the reduction of the CGO problem to an offline RL problem. While the overall template of the proof will naturally follow the proof structure of the underlying base offline RL analysis, our proof is based on the **novelly designed augmented MDP** (which is a goal-conditioned MDP). This step is critical, as **no existing proof can show the effectiveness of the offline goal-conditioned RL if only positive data (context-goal pairs in our case) is available**, which is our offline CGO setup. In order to prove our results, **we use a careful reformulation of the Bellman equation of the augmented MDP and construct an augmented value function and policy class in the analysis using the CGO structure**. As such, we respectfully disagree that the proof is trivial.
>
> ### 3. Not adequate experiments:
> Could you specify the reasons? In our environments, we use different dynamics datasets in different mazes, and for each maze we have three different context-goal setups representing the different CGO setups in the spectrum discussed in **Section 6 (L265)**. We show that our method works well and consistently outperforms the baseline methods in these different scenarios. Moreover, the main goal of this paper is to formulate the CGO setup since it is not formally studied in the literature, and as the first milestone to show it can be solved with commonly available offline datasets, which the proposed method demonstrates.
>
> ### 4. Typos:
> Thanks for pointing them out, and we will fix them in the revised draft.
>
> We hope our response resolves all the concerns in your review and please feel free to let us know if you have any other questions in the discussion period. If our responses have addressed your concerns, please kindly consider raising the score. Thank you!

---

> ### Author Response · Authors · 2024-08-12
>
> Dear reviewer, thank you again for the review and we hope our rebuttal addresses your concerns. We would greatly appreciate your feedback and please feel free to let us know if you have any other questions in the discussion period!

---

### Official Review · Reviewer_CzDa · 2024-07-16

**Soundness:** 3
**Presentation:** 3
**Contribution:** 3
**Rating:** 7
**Confidence:** 2

**Summary:**

This paper proposes a simple action-augmented MDP formulation for contextual goal-oriented problems in an offline RL setting. They show that their action-augmented MDP has a regret that is equivalent to the original MDP and any policy can be converted interchangeably without changing the regret. Along with theoretical justification, they also show that better performance compared to other reward prediction and goal-oriented RL methods in the experiments.

**Strengths:**

- The paper introduces and solves an interesting and challenging problem of context-defined goal-oriented RL policy learning, and do so without having access to labeled samples for the task.
- The authors propose a very effective and clever solution for converting existing dynamics and context-goal datasets to first create learning data, and then create augmented MDP policies while establishing learnable guarantees using only positive data (Th 5.4).
- The empirical performance is strong and generally outperforms other baselines.

**Weaknesses:**

- I am unsure how well the theoretical assumptions and setting of the paper transfers to more real world and larger-scale environments.

- The empirical evaluation has been only performed using a single environment, so the generalizability of the method to more diverse settings is unclear.

**Questions:**

- I am curious as to why is there a wide performance variance in Tables 1,2,3 between different env or method. Specifically, why is CODA significantly outperforming baselines in few settings while only providing marginal benefits in few others.

- The poor performance of goal-oriented RL methods in basic setting in Table 1 is also surprising, I request the authors to add more insights into this either in the rebuttal or in the final version.

**Limitations:**

The significant limitations have been delineated well in Sec 6.4.

---

> ### Author Rebuttal · Authors · 2024-08-06
>
> We sincerely appreciate your valuable comments. We found them extremely helpful in improving our draft. We address each comment in detail below.
>
> ### 1. “How well the theoretical assumptions and setting of the paper transfers to real world”:
>
> For the assumptions, our assumption **5.1** and **5.2** are general assumptions about the expressiveness of the value functions which are standard in offline RL theories. In plain words, we need the value function approximators to be expressive enough to effectively perform approximate dynamic programming in offline RL.
>
> For the offline CGO setup, we assume a dataset of dynamics data and a dataset of context-goal examples, which are **commonly available**. Notice the dynamics data is task agnostic and the context-goal examples do not require expert demonstrations. For example, in the example in Introduction where we instruct the truck to feasible warehouses, we only need the logged data of truck driving without any labels (dynamics data), and a set of instructions (contexts) and the corresponding warehouses (goals). We don’t need the expert trajectories of how the trucks can be navigated under different instructions (contexts) which allows commonly available dataset to be used in our setup.
>
> ### 2. “The generalizability of the method to more diverse settings is unclear”:
> While all the experiments are done using the AntMaze simulators, we specifically design each experiment to reflect the **diverse scenarios that offline CGO problems can cover**. In **Section 6 (L265)** and **Fig 2**, we propose a taxonomy of offline CGO problems based on context-goal relationship with an increasing complexity, and we design experiments to cover all these scenarios. Moreover, for each scenario, we use various dynamics datasets and different mazes. The experimental results show the proposed algorithm can consistently solve all these different scenarios.
> We used the same AntMaze domain throughout these experiments so that we can study – through controlled experiments – the effects due to different data relationships in different CGO settings. If we were to implement one setup from Fig 2 in one domain and another setup from Fig 2 in another one, there may be additional confounders due to the domain change. We hope these results provide good evidence to show the generalizability of our method to different CGO settings. Nonetheless, we acknowledge experiments with more realistic datasets and environments would be interesting future work.
>
> ### 3. “Why is there a wide performance variance”:
> We want to first highlight that **Table 1, 2, 3** are from **different context-goal relationships**. Despite being implemented via the same simulator (of the same dynamics), they mean totally different learning problems, so they should not be directly compared. We should only compare algorithms within the same table, not across tables.
> For each table, we provide the results of training with oracle reward, which is the skyline reference performance with respect to each dynamics dataset. The difference between that skyline and the other baselines is the room for improvement. Sometimes the headroom can be small, e.g., some coverage assumption is met like all goals in the dataset are feasible or the context-state reward can be easily learned from the dataset. In these tables we see that our proposed approach gives consistent improvement over baselines whenever possible. We will better clarify how the tables should be interpreted in the revision.
>
>
> ### 4. “Poor performance of goal prediction baseline in Table 1”:
> The reason is that as shown in **Figure 2(a)**, the goal area for this setup is very small, thus the number of goal examples in the context-goal dataset is very limited, thus providing difficulty for learning the goal distribution given context for goal prediction methods since it needs to learn to generate the goals first and then sample the goal as the condition.
>
> We hope our response resolves all the concerns in your review and please feel free to let us know if you have any other questions in the discussion period.

---

> > ### Comment · Reviewer_CzDa · 2024-08-08
> > **Thanks for the response**
> >
> > I thank the authors for taking time and drafting the response to my queries, I will keep my rating of accept. Also, I am really sorry at the set of reviews you got for this submission, it is indeed frustrating when avg reviewer confidence is 2.25/5 which only shows severe mismatch between the reviewer assignments and expertise. I wish you better luck at least in a next venue.

---

### Decision · Program_Chairs · 2024-09-25

**Decision:**

Accept (poster)

**Comment:**

The paper received the following scores: R1: 7 - Accept , R2: 4 - Borderline reject, R3: 5 - Borderline accept
R4: 7 - Accept (with R4 being the least credible and should be discarded).

Overall, reviewers highlight the interesting and challenging problem statement, the fact that the paper presents a general solution, as well as the good performance on empirical evaluations.

On the negative side, especially R2 points out that there is limited novelty, that the technique itself is straightforward, as well as limited experimentation on a very simple problem set.

Overall, I tend to agree with R2 for the following reasons: While the idea is interesting and designed for a general setup, the evaluation is limited to a single scenario. So, the evaluation does not really help to understand how generalizable the setup is, e.g., by evaluating a second scenario.

Usually, I would highly recommend the authors to improve the paper in this direction and extend the evaluation, probably in supplementary, to more setups and let the paper pass. In this case, the authors clearly stated that they do not intend to do this. We therefore had an intense discussion including SACs, and the decision was made to accept the paper. I hope that authors take the advice of R2 and AC serious, and strive for an improved CR version to strengthen their submission.